# ContraLog: Log File Anomaly Detection with Contrastive Learning and Masked Language Modeling

## Abstract

Log files record computational events that reflect system state and behavior, making them a primary source of operational insights in modern computer systems. Automated anomaly detection on logs is therefore critical, yet most established methods rely on log parsers that collapse messages into discrete templates. This discretization discards valuable information. Variable log values are ignored, semantic variation is lost. We propose ContraLog, a parser-free and self-supervised method that reframes log anomaly detection as predicting continuous message embeddings rather than discrete template IDs. ContraLog combines a message encoder that produces rich embeddings for individual log messages with a sequence encoder to model temporal dependencies across sequences. ContraLog is trained with a combination of masked language modeling and contrastive learning to predict masked message embeddings based on the surrounding context. Experiments on the HDFS, BGL, and Thunderbird benchmark datasets empirically demonstrate ContraLogs effectiveness on complex datasets with diverse log messages. Additionally, we find that message embeddings generated by ContraLog carry meaningful information and are predictive of anomalies even without sequence context. These results highlight embedding-level prediction as an approach for log anomaly detection, with potential applicability to other event sequences such as IoT telemetry and audit trails.

## 1 Introduction

Log files contain human-readable text snippets that record runtime events reflecting the internal state of a system. They often capture various computational events ranging from configuration changes and user requests to error messages. These logs are the primary record of past system activity and are often the main source of information for remotely operated systems. For security-critical applications, sufficient logging can even be required by regulatory guidelines such as the Payment Card Industry Data Security Standard (PCI Security Standards Council, 2022).

As systems grow in size, so does the number of generated logs, some producing millions of logs per minute (Mi et al., 2013). The volume, complexity, and diversity of these logs make manual assessment impractical, thereby increasing the need for robust and automated anomaly detection systems. Such automated approaches are important for the timely identification of abnormal events that could signal unauthorized access, performance issues, or system failure.

Anomalies in log files can manifest in various forms, including point anomalies, contextual anomalies, and collective anomalies (Landauer et al., 2023). Point anomalies are individual log entries that deviate significantly from the norm, while contextual anomalies are log entries that appear normal in isolation but are anomalous in the context they appear in. Collective anomalies represent abnormal deviations of entire log sequences.

Since anomaly labels are rarely available in real-world logs (He et al., 2016), we use a self-supervised method that models normal behavior and detects deviations without relying on labeled anomalies.

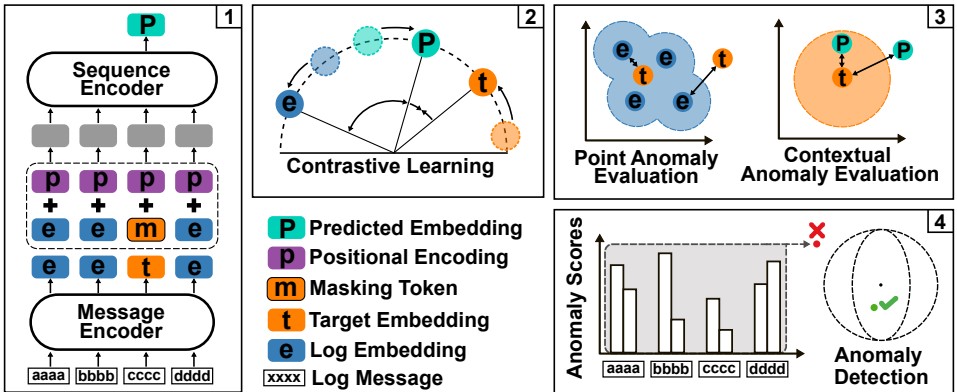

Figure 1: Overview of ContraLog. (1) Log messages of a sequence are individually embedded. Then, random embeddings are masked, and the SequenceEncoder predicts the original masked representations based on the context they occur in. (2) Both encoders are trained end-to-end using contrastive learning. We aim to maximize the similarity between the observed and the predicted embedding. (3) Anomalies are evaluated by two criteria. The point anomaly score estimates how similar a new embedding is to those seen during training. The contextual anomaly score measures how close the predicted embedding is to the observed one. (4) At inference, each message in a sequence is masked one after another, resulting in two anomaly scores for each message. For anomaly detection, we aggregate the scores into sequence-level features, which are standardized with robust z-scores and combined via the $L_2$ norm into a single anomaly score.

Traditional anomaly detection methods, whether based on classical statistical techniques or deep learning, often rely on log parsers such as Drain (He et al., 2017) or Spell (Du & Li, 2016) to organize log messages into a more structured format. Log files are typically generated using a consistent pattern, where each entry consists of a fixed message template (log key) and variable log values that depend on the context in which the message is recorded. Log parsers take advantage of this pattern by estimating the template used for individual messages based on a dataset of log entries.

These parsers are used in log anomaly detection because they provide a structured and simplified representation of log messages. By converting raw logs into a discrete set of unique templates, log parsers reduce the complexity of the data and make it more manageable for machine learning models.

Despite their widespread use, log parsers introduce several challenges:
**Information Loss**: Regardless of the log values, messages of the same type are represented by the same key, leading to a potential loss of information. For example, a voltage reading of 10 V is not equal to 100 V, but log parsers may treat them identically if the same template was used to generate logs for both measurements.
**Parsing Errors**: Parsers require dataset-specific rules and frequent updates as log schemas evolve.
**Semantic Similarity**: Templates with different IDs may be semantically similar, yet such similarity is ignored after discretization.

In short, log anomaly detection has been treated as a discrete prediction problem, even though log events are rich in continuous information. This approach leads to methods that ignore variations where anomalies can occur. To address these limitations, we ask:

> Can log anomaly detection be reframed from predicting discrete tokens to predicting continuous embeddings of raw messages?

To this end, we propose ContraLog, a novel approach to log file anomaly detection that combines contrastive learning with masked language modeling to work directly on raw log messages. ContraLog consists of a MessageEncoder that encodes individual log messages and a SequenceEncoder to capture sequential patterns within a sequence of message embeddings. This hierarchical design additionally shortens the effective sequence length compared to concatenating entire log messages

into one sequence. By training the model to predict masked message embeddings from the context they occur in, ContraLog learns to generate meaningful representations of log messages.

We argue that if a model is capable of predicting the masked embedding based purely on the context in which it appears, the associated message is an expected one and thus normal. If the model fails to predict the message embedding, the log is unexpected and likely not normal in the context it appears in. Additionally, message embeddings that deviate from known normal messages can provide an additional signal for anomaly detection, regardless of context.

By focusing on continuous embeddings rather than discrete IDs, ContraLog retains semantic and variable log parameters. We find that on the BGL and Thunderbird datasets, these embeddings themselves already separate normal from abnormal messages, yielding strong predictive power even without context.

Our work makes the following main contributions:

- A parser-free and self-supervised approach to log file anomaly detection trained with contrastive learning and masked language modeling.
- A detailed evaluation of ContraLog on three datasets, demonstrating its effectiveness without dataset-specific preprocessing.
- Evidence that meaningful embeddings alone carry predictive power, particularly on BGL and Thunderbird.

## 2 RELATED WORK

**Rule-Based Methods:** Historically, methods for detecting unusual workloads and resource usage (Barham et al., 2004; Reynolds et al., 2006; Thereska & Ganger, 2008) have relied heavily on domain knowledge. These approaches can be highly effective when the types of anomalies are well understood, and experts can establish clear rule sets. However, manual rule creation is time-consuming and requires expertise with the target system, making it difficult to scale to large and evolving datasets.

To address these limitations, more automated and data-driven methods have been developed.

**Machine Learning Methods:** These methods often parse the messages and then analyze windows of logs. They count how often each message type appears in a window and use the counts to create a feature vector for each sequence. Alternatively, Term Frequency-Inverse Document Frequency (TF-IDF) (Aizawa, 2003) can be used to create a feature matrix. For this approach, each window can be treated as a document. Supervised learning techniques, such as Support Vector Machines (SVM) (Liang et al., 2007), Nearest Neighbor (Liang et al., 2007), and Decision Trees (Chen et al., 2004) have been employed for log file anomaly detection. These methods require labeled data to train models that can classify log messages as normal or abnormal. While supervised methods can achieve high accuracy, they are limited by the need for labeled data, which is often not available in real-world scenarios (He et al., 2016). Unsupervised learning techniques, such as Log Cluster (Lin et al., 2016) and Principal Component Analysis (PCA) (Xu et al., 2009), do not require labeled data. These methods identify anomalies based on the inherent structure of the log data. Log Cluster groups similar log messages together, while PCA reduces the dimensionality of the log features to more easily identify unusual log sequences. Other unsupervised approaches for log-based anomaly detection include Isolation Forest (Liu et al., 2008), One-Class SVM (OCSVM) (Schölkopf et al., 2001; Li et al., 2003), and $k$-nearest neighbors with automatically labeled samples (Ying et al., 2021).

**Deep Learning Methods:** Deep learning methods have shown promise in capturing the complex contextual and temporal dependencies found in log sequences (Landauer et al., 2023). Early approaches use recurrent neural networks to model the sequential nature of log data. For example, DeepLog (Du et al., 2017) employs a Long Short-Term Memory (LSTM) network to predict the next log message in a sequence. Deviations from the predicted sequence are flagged as anomalies. LogAnomaly (Meng et al., 2019) follows a similar strategy but enhances the model by encoding the semantic meaning of log templates, which helps generate more meaningful representations of the log data.

More recent methods rely on the transformer architecture (Vaswani et al., 2017) to capture contextual information. LogBERT (Guo et al., 2021), for instance, adopts a BERT-based model (Devlin et al., 2019) to learn contextual representations from parsed log messages. In this framework, the model is trained using self-supervised tasks, specifically masked log key prediction and volume of hypersphere minimization, which enable it to model the common patterns of normal log sequences. Rather than working on the actual raw log messages, LogBERT operates on log keys by assigning each template a learnable embedding. In contrast, LogFit (Almodovar et al., 2024) bypasses the parsing step and applies masked language modeling directly to the tokens of concatenated log sequences. In both cases, the prediction target is part of a discrete set. LogELECTRA (Yamanaka et al., 2024) focuses on the detection of point anomalies by estimating how likely each token in a message has been replaced.

An alternative direction is represented by supervised approaches such as NeuralLog (Le & Zhang, 2021), which embeds individual log messages using a pretrained BERT model and then feeds a sequence of representations into another transformer to classify them as normal or abnormal.

Self-supervised representation learning has increasingly favored predicting embeddings rather than reconstructing raw inputs. The Joint-Embedding Predictive Architecture (Assran et al., 2023) uses separate context and target encoders plus a predictor to align latent representations, demonstrating greater efficiency and robustness. In the text domain, SimCSE (Gao et al., 2021) applies contrastive learning directly at the sentence-embedding level, producing semantically rich representations. However, log anomaly detection has so far focused primarily on discrete template prediction or raw message reconstruction, omitting the potential benefits of embedding-level prediction.

## 3 METHODS

### 3.1 MODEL

ContraLog[1] consists of three main components, a byte pair encoding (BPE) tokenizer (Sennrich et al., 2016), along with the MessageEncoder, and the SequenceEncoder, both of which are transformer encoders.

**Tokenizer:** Log messages generated with the same template usually share common phrases, resulting in a repetitive text corpus that lacks the variability typically found in most natural language datasets. This characteristic presents a challenge for many pretrained tokenizers, which are typically trained on diverse natural language corpora. As a result, log messages are often split into numerous fine-grained tokens, leading to inefficiencies. Moreover, log messages generally have a less diverse vocabulary compared to natural text, causing many tokens from pretrained tokenizers to occur rarely or not at all. To address these issues, we fit a new byte pair encoding tokenizer (Sennrich et al., 2016) to each dataset individually. This approach effectively compresses repetitive parts of log message templates into fewer tokens, reducing the number of input tokens and the required vocabulary size (see Figure 3 in the Appendix). Importantly, we do not pretokenize messages on spaces, allowing the tokenizer to build long tokens that can represent larger parts of log templates.

**MessageEncoder:** After tokenization, each log message is processed by a transformer encoder, referred to as the MessageEncoder. This step converts individual log messages into corresponding representations. Each tokenized log message $X_i$ is encoded as $T_i = \text{MessageEncoder}(X_i)$, where $T_i \in \mathbb{R}^{l \times d}$ with $l$ as the length of the token sequence and $d$ as the dimensionality of the representation space. $i \in 1, ..., n$ refers to the position of a message in a chronologically ordered sequence of $n$ total messages. The MessageEncoder also incorporates a learnable positional encoding to retain token order within each message. To obtain a single representation $E_i$ for each log message, we perform mean pooling over each token sequence $T_i$, followed by a linear layer, such that $E_i = \text{Linear}(\text{MeanPool}(T_i))$, with $E_i \in \mathbb{R}^d$.

**SequenceEncoder:** The sequence of log message representations is then added to another set of fixed positional encodings and fed into a second transformer encoder, which we refer to as the

---

[1] https://anonymous.4open.science/r/ContraLog-anonymous-repo-130E

SequenceEncoder. This second encoder captures the temporal dependencies and contextual information across the sequence of log messages. Outputs from the SequenceEncoder are then used to train both encoders via a variation of masked language modeling as described in section 3.2.

## 3.2 TRAINING

Unlike established parser-based approaches which predict a probability distribution over a set of discrete log keys, ContraLog encodes log messages into continuous embeddings. To train the MessageEncoder and SequenceEncoder, we therefore rely on a combination of masked language modeling and contrastive learning with a version of the InfoNCE loss (Oord et al., 2018). Let $|M|$ be the number of masked messages in a minibatch and index them by $j, i \in \{1, \ldots, |M|\}$. For each masked position $j$ the SequenceEncoder predicts $\hat{E}_j \in \mathbb{R}^d$ and the MessageEncoder provides target embeddings $E_i \in \mathbb{R}^d$, which are both normalized to unit length. We form the similarity matrix $S \in \mathbb{R}^{|M| \times |M|}$ with $S_{j,i} = sim(\hat{E}_j, E_i) = (\hat{E}_j^T E_i)/(||\hat{E}_j|| \, ||E_i||)/\tau$ with $\tau$ as a temperature parameter.

We define the row-wise cross-entropy loss

$$\mathcal{L}_{\text{row}}(S) = -\frac{1}{|M|} \sum_{j=1}^{|M|} \log \frac{\exp(S_{j,j})}{\sum_{k=1}^{|M|} \exp(S_{j,k})} \,, \tag{1}$$

which treats the diagonal element $S_{j,j}$ as the positive logit (or similarity) for row $j$. Analogously, define the column-wise cross-entropy loss by applying the same formula to the transpose $S^\top$:

$$\mathcal{L}_{\text{col}}(S) = \mathcal{L}_{\text{row}}(S^\top) = -\frac{1}{|M|} \sum_{i=1}^{|M|} \log \frac{\exp(S_{i,i})}{\sum_{k=1}^{|M|} \exp(S_{k,i})} \,. \tag{2}$$

Finally, the symmetric loss used for training is the average of the two directions:

$$\mathcal{L}_{\text{sym}} = \frac{1}{2}\big(\mathcal{L}_{\text{row}}(S) + \mathcal{L}_{\text{col}}(S)\big) \,. \tag{3}$$

## 3.3 INFERENCE

During inference, our method applies the learned MessageEncoder and SequenceEncoder to detect anomalies in log sequences by combining both contextual and point anomaly detection.

**Contextual Anomaly Detection:** For each masked log message in a sequence, the SequenceEncoder predicts a representation $\hat{E}$ from the context it occurs in. A high dissimilarity between the predicted and actual representations indicates a message not seen during training or one that is unexpected given the context in which it occurs, and thus a potential anomaly. We define an anomaly score for each log message as $\text{ContextScore}(E_j) = 1 - \text{sim}(\hat{E}_j, E_j)$.

To compute an anomaly score for the entire sequence $S$, we mask each of the $n$ messages one by one and perform one forward pass. This results in a contextual anomaly score for each message. We then aggregate these scores to obtain a single score for the entire sequence, either by taking the maximum or the mean of the individual scores

$$\text{SequenceScore}_{\text{max}}^{\text{context}}(S) = \max_{j \in \{1, \ldots, n\}} \text{ContextScore}(E_j) \tag{4}$$

$$\text{SequenceScore}_{\text{mean}}^{\text{context}}(S) = \frac{1}{n} \sum_{j=1}^{n} \text{ContextScore}(E_j) \,. \tag{5}$$

**Point Anomaly Detection:** While contextual anomaly detection captures temporal and contextual dependencies within log sequences, it has limitations. Specifically, when a sequence consists of entirely identical messages, the model can reasonably assume that the masked out message also contains the same message. To fill the mask, the SequenceEncoder can then reproduce the message representations of the other messages in the sequence. This issue can arise even with message types

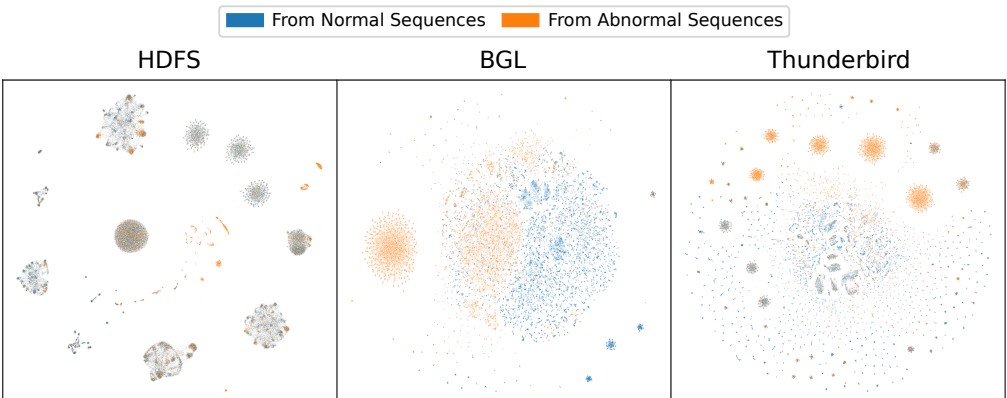

Figure 2: Visualization of the embedding spaces for messages from the HDFS, BGL, and Thunderbird datasets using UMAP dimensionality reduction (McInnes et al., 2018). Orange embeddings originate from abnormal sequences and blue embeddings from normal sequences.

not seen during training, potentially leading to false negatives where abnormal messages are not detected due to the context in which they occur.

To address this limitation, we introduce point anomaly detection. This method operates directly on the embeddings generated by the MessageEncoder, focusing on the intrinsic properties of individual log messages rather than their context. Looking at the embedding space of individual log messages (see Figure 2), we can see a partial separation between the embeddings of log messages from normal and abnormal sequences. By fitting an anomaly detection model to a subset of embeddings from normal sequences, we can detect messages that deviate from the expected distribution in the embedding space. To detect outliers, we calculate the distance to the closest embedding seen in a random subset of the training set.

The distance for embedding $E_j$ is defined as $\mathrm{PointScore}(E_j) = 1 - \mathrm{sim}(E_j, E_{\mathrm{nearest}})$, $E_{\mathrm{nearest}}$ is the closest embedding in the subset of normal training sequences. To aggregate scores for a sequence, we calculate the mean and maximum of these scores

$$\mathrm{SequenceScore}_{\mathrm{max}}^{\mathrm{point}}(S) = \max_{j \in \{1, \ldots, n\}} \mathrm{PointScore}(E_j) \tag{6}$$

$$\mathrm{SequenceScore}_{\mathrm{mean}}^{\mathrm{point}}(S) = \frac{1}{n} \sum_{j=1}^{n} \mathrm{PointScore}(E_j) . \tag{7}$$

### 3.4 ANOMALY DETECTION

Let $Y_{\mathrm{cal}} \in \mathbb{R}^{m \times 4}$ be the matrix of sequence-level scores from normal sequences used for calibration, where each row corresponds to a sequence and each column to one of the four SequenceScores (mean and maximum, point and context scores).

For each feature $f \in \{1, 2, 3, 4\}$ we compute the median and the median absolute deviation (MAD) on $Y_{\mathrm{cal}}$:

$$\tilde{\mu}_f = \mathrm{median}\big((Y_{\mathrm{cal}})_{:,f}\big), \qquad \mathrm{MAD}_f = \mathrm{median}\big(|(Y_{\mathrm{cal}})_{:,f} - \tilde{\mu}_f|\big) . \tag{8}$$

The feature-wise robust z-score for sequence $s$ is

$$\mathrm{rz}_{s,f} = \frac{\big|y_{s,f} - \tilde{\mu}_f\big|}{\max\big(\mathrm{MAD}_f, \varepsilon\big)} , \tag{9}$$

where $y_{s,f}$ denotes the $f$-th score of sequence $s$ and $\varepsilon > 0$ is a small constant introduced to avoid division by zero. The calibration quantities $\tilde{\mu}_f$ and $\mathrm{MAD}_f$ are computed once on $Y_{\mathrm{cal}}$ and stored.

We aggregate the four feature-wise robust z-scores for each sequence $s$ using the $L_2$ norm:

$$\text{score}_s \;=\; \sqrt{\sum_{f=1}^{4} \text{rz}_{s,f}^2} \; . \tag{10}$$

The detection threshold $\theta$ is set to the empirical 95th percentile of the calibrated scores computed on $Y_{\text{cal}}$. At inference, $\text{rz}_{s,f}$ and $\text{score}_s$ for new sequences are computed using the stored $\tilde{\mu}_f$, $\text{MAD}_f$ and $\theta$. A sequence is classified as anomalous if $\text{score}_s > \theta$. Appendix A.2 gives further details on threshold selection.

By combining both contextual and point anomaly detection, our approach captures temporal dependencies and intrinsic message-level deviations. The mean anomaly scores enables the model to capture collective anomalies. For these anomalies log message in a sequence may appear normal when considered in isolation, but abnormal when considered collectively, as described by Ruff et al. (2021). The relative magnitude of the feature-wise robust z-scores indicates which anomaly score contributes most to the overall detection. This allows us to characterize whether an anomaly is primarily contextual, point-based, or a mixture of both. A more detailed analysis of feature contributions is provided in Appendix A.3.

Some messages are exactly repeated throughout the datasets multiple times (see Table 1). During inference, we can prevent redundant calculations by the MessageEncoder by building a dictionary of embeddings with the message text as a key. Once a message has been processed, its embedding is stored and reused for subsequent occurrences, effectively skipping the embedding step. For the Thunderbird dataset, caching can reduce embedding steps by up to 89.1%. Appendix A.4 provides more details on the effects of caching.

## 4 EXPERIMENTS

To evaluate the effectiveness of our approach, we compared the ability of ContraLog in detecting abnormal log sequences to that of several baseline methods:

**LogBERT:** A transformer-based model that learns contextual representations via masked language modeling. Unlike ContraLog, LogBERT requires log parsers to structure the input data and uses a more classical masked language modeling approach to predict the probability of a masked log belonging to a certain log template.

**DeepLog:** An LSTM-based approach that models sequential log behavior by predicting the next log event. DeepLog also depends on log parsers to extract event sequences and flags anomalies when predictions do not match observed logs.

**Statistical Machine Learning:** Methods such as One-Class SVM (OCSVM) and Isolation Forest are applied on count vectors that record the frequency of each log message type in a sequence. These features ignore the order and temporal dependencies of messages, detecting anomalies based solely on deviations from expected log key distributions.

**Data** We evaluate all methods on three datasets: HDFS, BGL, and the full Thunderbird dataset (Xu et al., 2009; Oliner & Stearley, 2007), which provide a wide range of sequence lengths, log types, and complexity levels. Sources for all these datasets can be found on LogHub (Zhu et al., 2023). For more detailed information about the number of sequences and sequence length, see Table 1 and the Data section in the Appendix A.5.

**Preprocessing** For methods that rely on log IDs, we adhere to the parsing parameters established by LogBERT, including dataset-specific regular expressions, tree depth, and similarity threshold. The regular expressions remove block IDs for messages from the HDFS dataset, hexadecimal numbers for the BGL dataset, and specific warning patterns for the Thunderbird dataset. Parsers are fitted on the entire dataset to avoid out-of-vocabulary issues, as highlighted in previous work by Almodovar et al. (2024). Additionally, we allow both methods to discard sequences shorter than ten messages. Other than in the original implementations, we split datasets chronologically with non overlapping sliding windows. Abnormal sequences from the training set are removed to prevent data leakage.

For the statistical machine learning baselines, feature vectors are built by counting how often each log key appears in a sequence. This is similar to a bag-of-words approach and ignores the order in which messages occur in, losing temporal information.

ContraLog requires minimal preprocessing. The only step we take is the extraction of the core message from the entire log line, ignoring metadata such as log creation timestamps and labels.

All methods were evaluated on a held-out test set with an equal number of normal and abnormal sequences.

Table 1: Dataset statistics including the number of normal and abnormal log sequences, the average number of messages per sequence, the total and unique numbers of log messages (ratio in parentheses), and the number of unique log keys identified by a Drain parser. Not every message in the HDFS dataset is part of a labeled sequence, resulting in a mismatch between session count, average sequence length, and total message number. By filtering out the session identifier Block IDs from the HDFS messages, the number of unique messages could be reduced to 1.802.378, resulting in a ratio of just 16.1%.

| Dataset | Normal/ Abnormal Sessions | Average Sequence Length | Total/Unique Messages | Log Keys |
|---|---|---|---|---|
| HDFS | 558,223/16,838 | 18.2 | 11.2M/10.3M - (92.4%) | 16 |
| BGL | 51,667/6,203 | 81.0 | 4.7M/1.8M - (37.8%) | 155 |
| Thunderbird | 488,137/89,446 | 233.09 | 211.2M/23.0M - (10.9%) | 4282 |

## 5 RESULTS

Table 2: Performance of various models evaluated on the HDFS, BGL, and Thunderbird datasets. The models include ContraLog, LogBERT, DeepLog, One-Class SVM (OCSVM), and Isolation Forest.

| Model | HDFS | | | BGL | | | Thunderbird | | |
|---|---|---|---|---|---|---|---|---|---|
| | Precision | Recall | F1-Score | Precision | Recall | F1-Score | Precision | Recall | F1-Score |
| **ContraLog** | 93.88 | 74.57 | **83.12** | 94.68 | 99.13 | **96.86** | 95.03 | 9.84 | **97.38** |
| **LogBERT** | 98.80 | 64.90 | 78.34 | 87.62 | 92.73 | 90.10 | 90.56 | 94.86 | 92.66 |
| **DeepLog** | 92.21 | 64.05 | 75.59 | 91.56 | 79.32 | 85.00 | 89.78 | 94.02 | 91.85 |
| **OCSVM** | 50.07 | 99.16 | *66.54* | 57.03 | 66.88 | 61.56 | 51.26 | 54.13 | 52.66 |
| **Isolation Forest** | 66.80 | 65.92 | 66.36 | 88.89 | 00.96 | 01.90 | 00.00 | 00.00 | 00.00 |

Table 2 summarizes the performance of various models on detecting abnormal log sequences in a held-out test set. We evaluated five approaches: ContraLog, LogBERT, DeepLog, as well as statistical methods, OCSVM and Isolation Forest. LogBERT and DeepLog were assessed using their respective open-source implementations.

Although the performance of the statistical methods depends greatly on the dataset and selected hyperparameters (see Table 4 in the Appendix), they can perform comparably to deep learning methods on the HDFS dataset. This suggests that the sequential order of messages may not be critical for identifying some anomalies. Specifically, anomalies which are characterized by the frequency or absence of specific message types rather than their order. This finding is also supported by experiments described in Appendix A.11, where the reordering of messages within a sequence had a smaller impact on detection performance than the removal or addition of messages.

LogBERT and DeepLog can generally outperform the statistical baselines. However, their scores are sensitive to the evaluation setting, particularly regarding the fitting of the log parser (see Appendix A.6.2).

ContraLog achieves consistently high F1 scores on all datasets. The results show that predicting latent embeddings can be as effective as, or even preferable to, predicting the exact log key as it enables the model to capture semantic overlap and variations that are lost when messages are collapsed into discrete templates. For instance, in cases where two log types appear almost interchangeably, a

model that relies on latent embedding similarity can capture the overlap in meaning without having to decide on a unique log key to predict.

Figure 2 illustrates a projection of the learned message embeddings for normal and abnormal sequences. While many clusters contain messages from both normal and abnormal sequences, some clusters exclusively contain abnormal messages. Especially for the BGL and Thunderbird datasets, a large proportion of abnormal sequences contain messages from these clusters, allowing the proposed point anomaly detection to identify them effectively. This works to such a degree that the exclusion of contextual anomaly scores does not decrease performance on the Thunderbird dataset and even slightly improves performance on the BGL dataset (see ablation study in the Appendix A.7).

Additionally, we find that the MessageEncoder does perform a task comparable to that of a parser, as it groups related log messages together in the embedding space. However, other than purely parser-based methods, the MessageEncoder can assign different representations to logs with the same template, but different parameters. In other cases, the encoder assigns similar embeddings to messages with different templates, if they occur in similar contexts (see Appendix A.9 for more details).

An additional benefit of our hierarchical approach is its computational efficiency (see Appendix A.4). By processing individual log messages with the MessageEncoder and operating on a much shorter sequence of embeddings with the SequenceEncoder, our method reduces the effective sequence length. Since the computational complexity of transformer models scales quadratically with sequence length, this design offers efficiency gains over the alternative approach where messages from a sequence are concatenated and then processed.

Overall, the experimental results validate the core hypothesis that predicting latent embeddings can be an effective alternative to exact log key prediction. ContraLog consistently achieves high performance across multiple datasets, supporting its applicability in diverse log analysis scenarios.

## 6 CONCLUSION

We introduced ContraLog, a parser-free approach for log file anomaly detection trained with contrastive learning and masked language modeling. By operating directly on raw log messages, ContraLog avoids the challenges associated with traditional parser-based methods. Our approach instead focuses on predicting continuous message embeddings, capturing semantic information and temporal dependencies of normal logs.

A central contribution of ContraLog is its two-pronged anomaly detection strategy, which combines contextual and point-based scoring. We show that especially point anomaly detection contributes to the overall performance on the BGL and Thunderbird datasets.

Another key advantage of ContraLog lies in its hierarchical processing strategy that keeps the input sequence length for the MessageEncoder and SequenceEncoder short. This architecture also enables efficient reuse of message embeddings through caching.The use of custom tokenizers furthermore reduce input length when compared to pretrained tokenizers.

Experimental evaluations on the HDFS, BGL, and Thunderbird datasets demonstrate that ContraLog achieves competitive performance across diverse logging scenarios. In particular, our method consistently attains high F1-scores on complex datasets, thereby validating the effectiveness of predicting latent embeddings over explicit log key prediction.

Overall, the combination of masked language modeling and contrastive learning in our approach yields meaningful representations that improve anomaly detection. By enabling early detection of deviations in system behavior, our method can support predictive maintenance, strengthen system security by flagging suspicious activity, and ultimately help to reduce downtime in critical infrastructure.

## 7 REPRODUCIBILITY

We provide a reference implementation for all components required to reproduce the main results. This includes code for data processing and labeling, model definitions, training and evaluation

scripts, as well as configuration files. Key hyperparameters and computational requirements are listed in Appendix A.10. Labeling rules are described in Appendix A.5. Used datasets are publicly available.

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

## A  APPENDIX

### A.1  TOKENIZER

Figure 3 visualizes the fitting behavior of our BPE tokenizer for different datasets. As the dictionary size increases, the average number of tokens required to tokenize a log message decreases. For this comparison, only the number of tokens that occur at least once in the corresponding dataset is displayed. The BPE tokenizer starts with a base dictionary of Unicode characters and merges tokens until the unpruned dictionary size reaches 4,096. For comparison, the unpruned dictionary of the BERT-Base Uncased tokenizer contains more than 30,000 tokens.

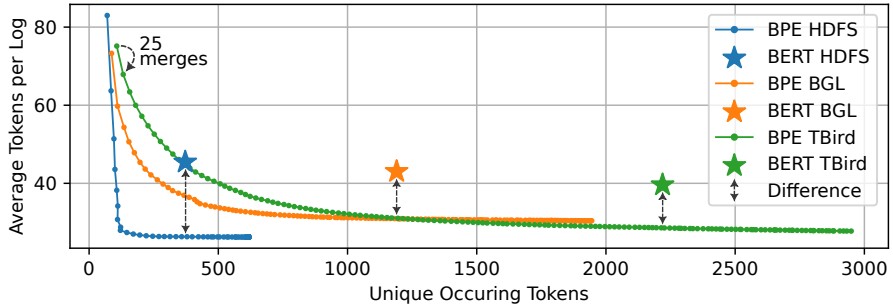

Figure 3: Comparison of log message compression rates between a custom Byte Pair Encoding (BPE) tokenizer and the pre-fitted BERT-Base Uncased (Devlin et al., 2019) tokenizer across different datasets.

**Tokenizer Generalization**  Log files are often highly specific to the software that generates them, characterized by repetitive structures and limited semantic complexity compared to general text data. As a result, patterns learned from one dataset may not transfer effectively to an unrelated dataset.

This uniqueness also impacts the tokenizer. Tokenizers fitted to one dataset often perform poorly when applied to another, leading to suboptimal tokenization. Table 3 shows the average number of tokens required to tokenize messages when using a tokenizer fitted on one dataset to process messages from another.

Table 3: Average number of tokens required for tokenization across datasets.

| Tokenizer Data | Message Data | Avg. Tokens |
|---|---|---|
| HDFS | HDFS | 20.81 |
| BGL | BGL | 24.88 |
| HDFS | BGL | 53.96 |
| BGL | HDFS | 54.04 |

### A.2  THRESHOLD SENSITIVITY ANALYSIS

While we report results using a 95th percentile threshold for all datasets, using the label information in the test set we can analyze how performance depends on the chosen threshold. Figure 4 shows the F1 scores achieved on the test set when using different percentiles of the normal anomaly score distribution on the training set as a threshold. Metrics were calculated using the full feature set of mean and maximum, point and context anomaly scores.

In practice, the optimal threshold depends on the distribution shift between normal and abnormal logs and the contamination of the training set with abnormal logs. If labeled information is available, the choice of threshold can have a significant influence on the performance. The theoretically optimal F1 scores could be 97.07 for BGL with a threshold of 96.10, 83.94 for HDFS with a threshold of 97.86 and 97.81 for Thunderbird with a threshold of 96.70.

Figure 5 shows the Receiver Operating Characteristic (ROC) curves for the HDFS, BGL, and Thunderbird datasets. The unique shape of the curve for HDFS is caused by a subset of easily detectable

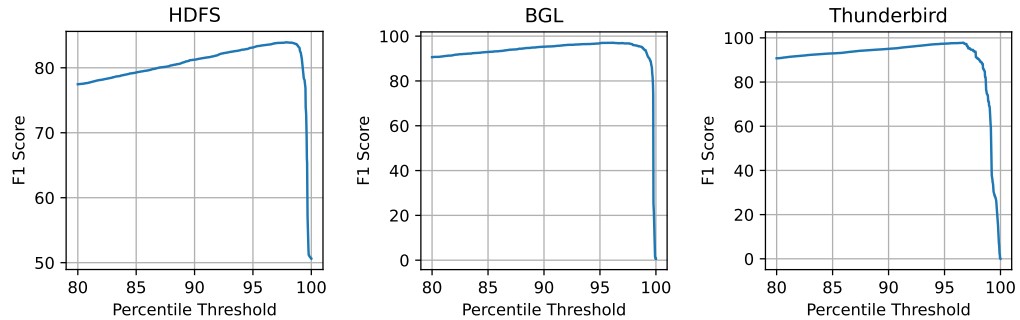

Figure 4: F1 scores achieved on the test set when using different percentiles of the anomaly score distribution on the training set as a threshold.

anomalies (short/long sequences, exceptions that only occur in abnormal sessions). This makes it possible to maintain a low false positive rate even with low thresholds.

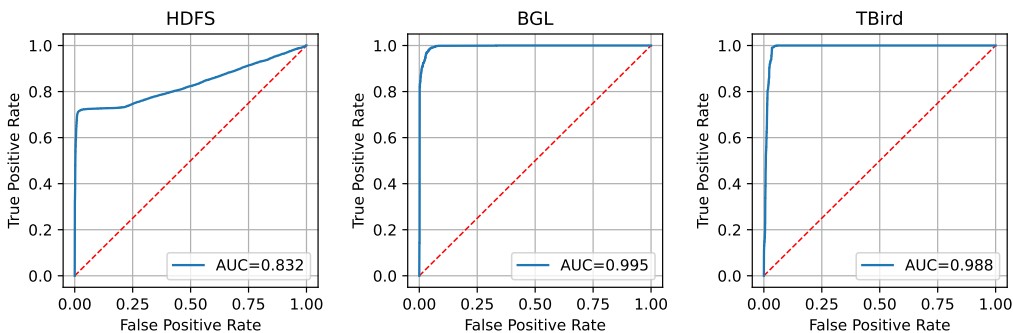

Figure 5: Receiver Operating Characteristic (ROC) curves for different datasets.

### A.3 SCORE FRACTIONS

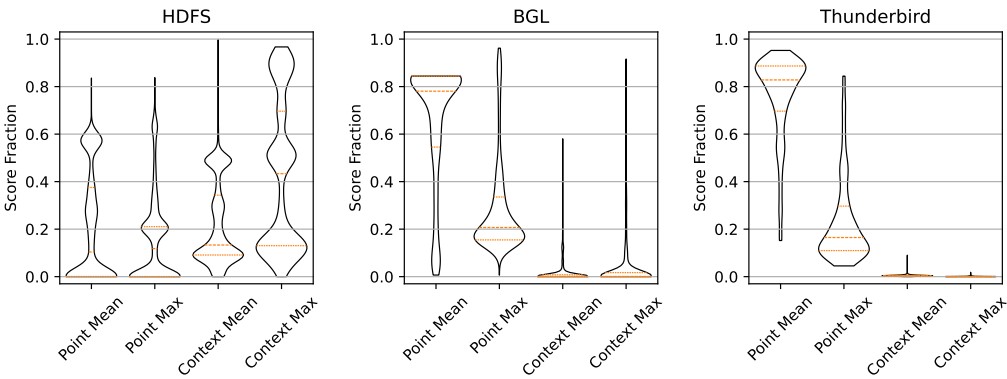

Figure 6: Distributions of the ratio different anomaly scores contribute to the classification of abnormal test sequences.

By normalizing the robust z-scores of a log sequence, we can estimate how much each anomaly score contributes to the final classification of a sequence. Figure 6 shows the distributions of the ratio different anomaly scores contribute to the classification. For different datasets, different anomaly scores hold different importance. While score contributions for HDFS are mostly balanced, BGL

and Thunderbird heavily rely on the mean and max point anomaly scores. Especially for the Thunderbird dataset, context anomaly scores play only a minor role. These observations align with the results of the ablation study in Appendix A.7. When detecting anomalies based on a single feature alone, the anomaly score with the highest average contribution per dataset achieves the best results.

## A.4 CACHE HIT RATES

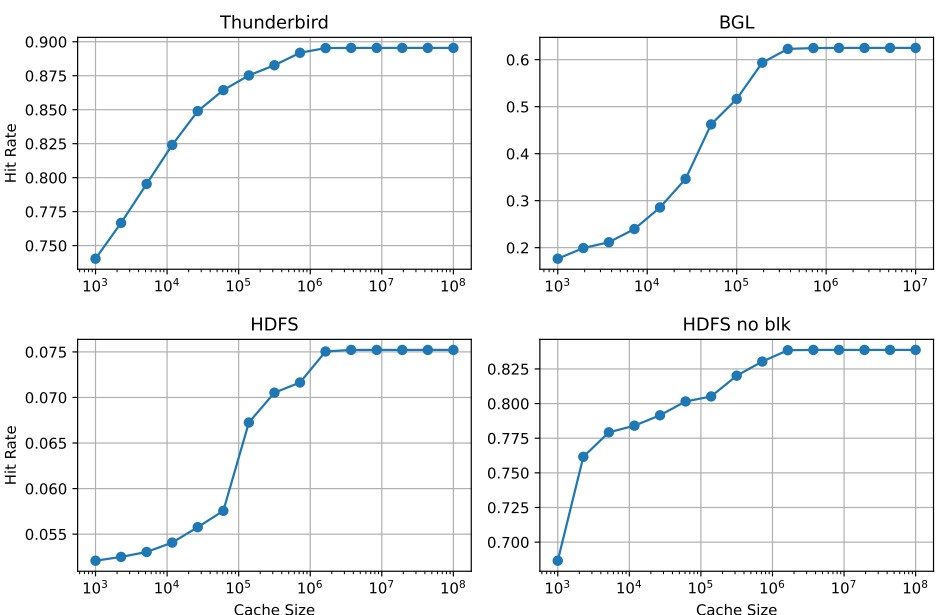

Figure 7: Cache hit rates for different configurations.

Figure 7 shows the log message cache hit rates for different cache sizes. For this experiment, we process datasets in chronological order. Two versions of the HDFS dataset were considered. One with the original unaltered messages (HDFS) and one with the session identifying block ids removed (HDFS no blk). Due to the repetitive nature of log messages, where identical messages appear multiple times, potentially in quick succession, caching of the LogEmbedder outputs can benefit even from small cache sizes. The cache hit rate is defined as the ratio of embeddings that were found in the cache (from previous identical messages) at the time the message appears in the dataset.

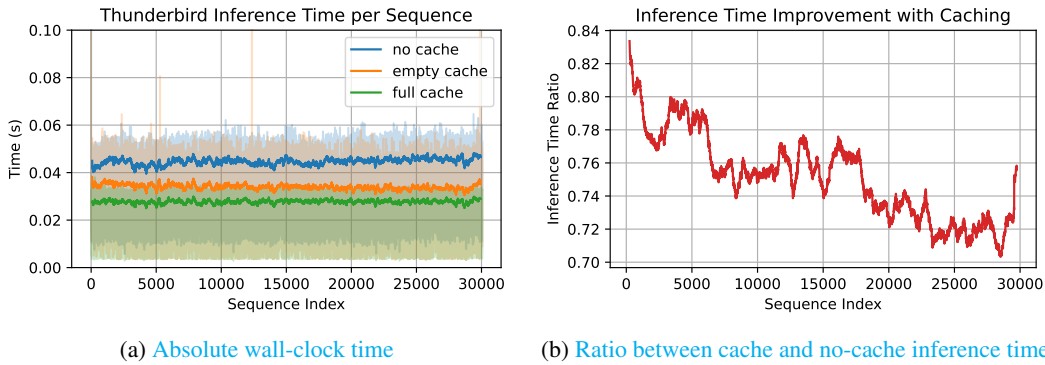

(a) Absolute wall-clock time      (b) Ratio between cache and no-cache inference time

Figure 8: Comparison of moving average inference timing on the first 30,000 Thunderbird sequences.

Figure 8 compares the absolute wall-clock time required for inference on the first 30,000 Thunderbird sequences with and without caching. For better clarity, the visualized metrics represent a

moving average over inference time for 500 sequences. Figure 8a shows the absolute time required to process each sequence for three caching configurations. 'No cache' represents the baseline with no caching. 'Empty cache' represents the scenario, where a new model starts with an empty cache, which is gradually filled during inference. Figure 8b Shows how the relative reduction in inference time (empty cache/no cache) improves as the cache fills up. The 'full cache' configuration represents the performance when iterating over the subset a second time. The average duration to calculate the contextual anomaly score for on thunderbird sequence is 0.045s without caching and 0.028s with a full cache.

## A.5   DATA

The HDFS, BGL, and Thunderbird datasets (Xu et al., 2009; Oliner & Stearley, 2007) provide a wide range of sequence lengths, log types, and complexity levels. Sources for all these datasets can be found on LogHub[2] (Zhu et al., 2023). For more detailed information about the number of sequences and sequence length, see Table 1.

**HDFS Dataset**   The HDFS dataset (Xu et al., 2009) consists of logs from the Hadoop Distributed File System. This dataset is organized into session windows, where each session contains a sequence of log messages. The dataset includes labels indicating whether each session is normal or abnormal. Since sessions are already clearly defined, we apply no further windowing. Instead, sessions longer than 256 messages are truncated while maintaining the original session label.

**BGL Dataset**   The BGL dataset (Oliner & Stearley, 2007) contains logs from the Blue Gene/L supercomputer. For this dataset, we used time windows of 60 seconds, capped to a maximum of 256 log messages per window. Each log message in the BGL dataset is labeled as normal or abnormal. If any message within a window is abnormal, the entire sequence is labeled as abnormal. This dataset is challenging because some windows are highly repetitive, with the same message appearing multiple times.

**Thunderbird Dataset**   The Thunderbird dataset (Oliner & Stearley, 2007) comprises logs from the Thunderbird supercomputer. Similar to the BGL dataset, we used time windows of 60 seconds, capped to a maximum of 256 log messages per window. The labeling works identically to the BGL Dataset.

The Thunderbird system creates more numerous and more complex logs compared to the other systems. The result is a higher average sequence length and a larger overall dataset. Furthermore, the structure of the log messages is more complex. Our Drain parsers (for more information, see Preprocessing in the Experiments Section 4) fitted on the training set was able to detect 4282 unique log types. Although the parsing parameters greatly influence the number of unique log IDs, this shows the difference in semantic complexity between the datasets.

This higher complexity creates a potentially more challenging environment for anomaly detection, although the overall difficulty also depends on the similarity between normal and abnormal sequences.

## A.6   BASELINES

### A.6.1   TUNED STATISTICAL MACHINE LEARNING METHODS

For the statistical machine learning methods, we note that the selected hyperparameters can greatly influence the performance (see Table 4). To give a better overview of the potential performance, we performed a grid search using a subset of normal and abnormal sequences.

In some cases, these statistical approaches perform comparably to deep learning methods, suggesting that for certain datasets, the sequential order of messages may not be critical for identifying anomalies. Count vectors also allow the easy detection of unusually short or long sequences, which can be an indicator of anomalies.

Still, the performance of ContraLogs remains better than that of the tuned statistical methods.

---

[2]https://github.com/logpai/loghub

In real-world settings, labeled data is often only available in limited quantities, making this type of hyperparameter tuning difficult.

Table 4: Performance of One-Class SVM (OCSVM), and Isolation Forest.

| Model | HDFS | | | BGL | | | Thunderbird | | |
|---|---|---|---|---|---|---|---|---|---|
| | Precision | Recall | F1-Score | Precision | Recall | F1-Score | Precision | Recall | F1-Score |
| OCSVM[T] | 86.81 | 72.55 | **79.03** | 69.35 | 86.35 | **76.92** | 80.34 | 93.57 | **86.46** |
| Isolation Forest[T] | 66.84 | 66.04 | 66.44 | 65.80 | 38.58 | 48.64 | 80.45 | 85.71 | 83.00 |

with [T] indicating experiments with tuned hyperparameters.

### A.6.2 DEEP LEARNING WITH ADJUSTED CONDITIONS

For Table 2 experimental conditions for LogBERT and DeepLog were adjusted to closely resemble those in the original implementations. For Table 5 experimental conditions for LogBERT and DeepLog were adjusted to match those used for ContraLog. By aligning these conditions, we observe a notable decrease in performance for both LogBERT and DeepLog:

- Parser is only fitted on the training set, not the whole dataset.
- Short sequences are not discarded.
- No overlap between sliding windows.
- No shuffling of sequences before splitting into train and test set.
- Only abnormal sequences originally in the test set are used for testing.

As noted by Le & Zhang (2022) model performance is highly influenced by the parsing step.

Table 5: Performance of LogBERT and DeepLog under modified conditions.

| Model | HDFS | | | BGL | | | Thunderbird | | |
|---|---|---|---|---|---|---|---|---|---|
| | Precision | Recall | F1-Score | Precision | Recall | F1-Score | Precision | Recall | F1-Score |
| LogBERT | 96.44 | 38.97 | 55.51 | 99.67 | 36.55 | 53.49 | 62.28 | 99.78 | 76.69 |
| DeepLog | 38.81 | 81.39 | 52.56 | 54.86 | 100.00 | 70.85 | 80.09 | 96.40 | 87.49 |

### A.7 SEQUENCESCORE ABLATION STUDY

Table 6: F1 score for anomaly detection with different combinations of anomaly scores. $P_{max}$ and $P_{mean}$ represent the max and mean point anomaly scores, and $C_{max}$ and $C_{mean}$ the max and mean context anomaly scores. Only features marked with a checkmark were used to compute the metrics in the respective row.

| $P_{max}$ | $P_{mean}$ | $C_{max}$ | $C_{mean}$ | HDFS | BGL | Thunderbird |
|---|---|---|---|---|---|---|
| - | - | - | ✓ | 79.317 | 36.172 | 38.799 |
| - | - | ✓ | - | 78.869 | 34.409 | 10.517 |
| - | - | ✓ | ✓ | 79.038 | 37.176 | 32.344 |
| - | ✓ | - | - | 69.871 | 92.288 | 96.344 |
| - | ✓ | - | ✓ | 82.364 | 93.267 | 96.354 |
| - | ✓ | ✓ | - | 81.600 | 94.643 | 96.344 |
| - | ✓ | ✓ | ✓ | 81.472 | 94.708 | 96.354 |
| ✓ | - | - | - | 69.577 | 97.281 | 97.513 |
| ✓ | - | - | ✓ | 82.016 | 97.425 | 97.513 |
| ✓ | - | ✓ | - | 81.305 | 96.907 | 97.513 |
| ✓ | - | ✓ | ✓ | 81.293 | 96.804 | 97.513 |
| ✓ | ✓ | - | - | 69.797 | 96.287 | 97.376 |
| ✓ | ✓ | - | ✓ | 82.382 | 96.380 | 97.376 |
| ✓ | ✓ | ✓ | - | 81.900 | 96.685 | 97.376 |
| ✓ | ✓ | ✓ | ✓ | 83.121 | 96.858 | 97.376 |

Table 6 presents the results of an ablation study examining the impact of different combinations of anomaly scores on the F1 score for anomaly detection. The study evaluates the importance of different anomaly scores across the HDFS, BGL, and Thunderbird datasets.

While combining all scores generally provides robust results, there are cases where a subset of scores performs equally well or better. The results show that the optimal combination of scores varies by dataset. The HDFS dataset achieves its highest F1 score (83.121) using all features, while the BGL dataset performs best (97.425) with $P_{max}$ and $C_{mean}$. In contrast, the Thunderbird dataset achieves its highest F1 scores (97.513) when $P_{max}$ but not $P_{mean}$ are included.

Without labeled data however, it is challenging to determine the optimal combination of scores for a given dataset. These observations align with the observations in Appendix A.3, where features that achieve a high F1 score on their own tend to also play an important role when combined with other features. Also, Wittkopp et al. (2021) found that most anomalies in the BGL and Thunderbird datasets can be considered as point anomalies, which aligns with our observations.

## A.8   POINT ANOMALY DETECTION REFERENCE SEQUENCES

Figure 9 shows the F1-scores for the BGL dataset when only relying on point anomaly scores for different amounts of reference sequences from the training set. To build the reference set, we randomly sample a subset of normal sequences from the training set and embed their messages. Since we calculate point anomaly scores based on the single nearest neighbor distance in the embedding space, duplicate messages can be discarded. For this experiment, the number of reference sequences is varied between 125 and 16,000, covering almost half of the entire training set.

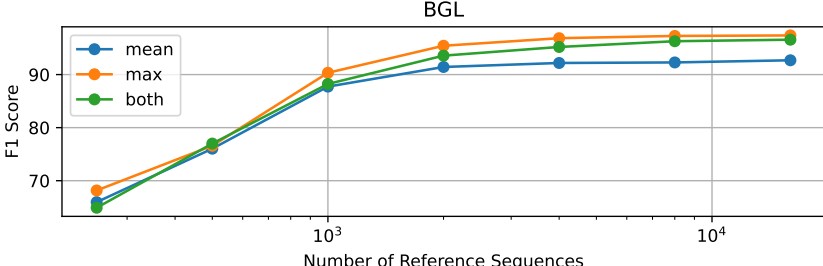

Figure 9: F1-scores for point anomaly detection on the BGL dataset when using different amounts of reference sequences from the training set.

## A.9   EMBEDDING SPACE ANALYSIS

Figure 10 visualizes the embedding space for HDFS log messages using a UMAP dimensionality reduction. Samples for this experiment were drawn from 1000 normal and 1000 abnormal sessions from the test set. To better understand how ContraLog groups messages, we manually analyze the embedding space. For the following analysis, variable log parameters are replaced with placeholders (e.g., <IP+Port>, <Block ID>). We find messages are approximately grouped into the following logical groups:

**A+B:** Messages from two different templates appear in this cluster. One set of messages about DataNodes serving blocks, e.g., *"INFO dfs.DataNode$DataXceiver: <IP+Port> Served block <Block ID> to <IP>"* and one set of messages about DataNodes encountering an exception while serving a block, e.g., *"WARN dfs.DataNode$DataXceiver: <IP+Port> :Got exception while serving <Block ID> to /<IP> :"*. Both messages appear in normal and abnormal sequences. Both clusters overlap, indicating the corresponding messages appear in similar contexts. Instead of a confirmation of the successful serving of a block, an exception might occur in its place.

**C:** Messages about the start of a block transfer, e.g., *"INFO dfs.DataNode: <IP+Port> Starting thread to transfer block <Block ID> to <IP+Port>"*. This group is split into multiple clusters, depending on the values of parameters. Variants with two target addresses (*"[...] <Block ID> to <IP+Port>, <IP+Port>"*) form their own cluster. Most messages of these cluster originate from abnormal sequences.

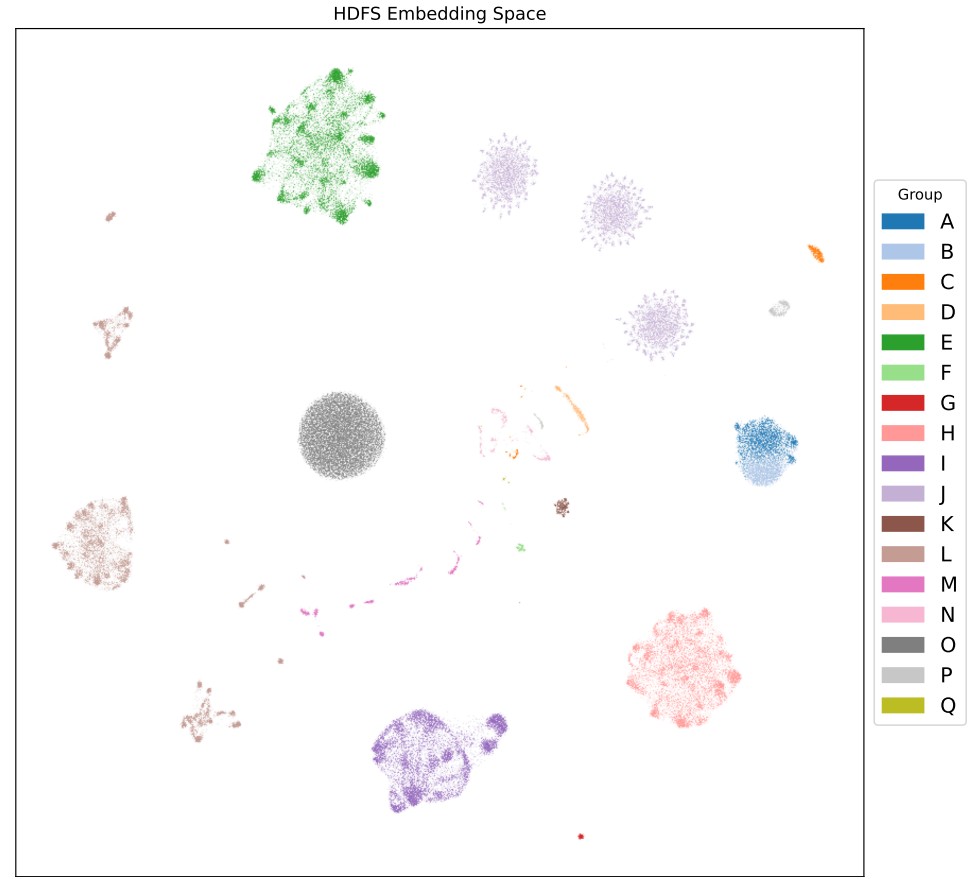

Figure 10: UMAP projection of the embedding space for messages from 2,000 HDFS sequences. Colors indicate manual groupings of messages with similar semantics.

**D:** Messages about the replication of blocks to other data nodes, e.g.,*"INFO dfs.FSNamesystem: BLOCK\* ask <IP+Port> to replicate <Block ID> to datanode(s) <IP+Port>"*.

**E:** Messages about updating block maps, e.g., *"INFO dfs.FSNamesystem: BLOCK\* NameSystem.addStoredBlock: blockMap updated: <IP+Port> is added to <Block ID> size <Block Size>"*.

**F:** Messages about blocks not belonging to any file, e.g., *"INFO dfs.FSNamesystem: BLOCK\* NameSystem.addStoredBlock: addStoredBlock request received for <Block ID> on <IP+Port> size <Block Size> But it does not belong to any file."*. All messages in this cluster only appear in abnormal sequences, making it suitable for point anomaly detection.

**G:** Messages about redundant requests, e.g., *"WARN dfs.FSNamesystem: BLOCK\* NameSystem.addStoredBlock: Redundant addStoredBlock request received for <IP> on <IP+Port> size <Block Size>"*. All messages of this cluster originate from abnormal sequences.

**H:** Messages about blocks being marked as invalid, e.g., *"INFO dfs.FSNamesystem: BLOCK\* NameSystem.delete: <Block ID> is added to invalidSet of <IP+Port>"*.

**I:** Messages about deleting blocks, e.g., *" INFO dfs.FSDataset: Deleting block <Block ID> file /mnt/hadoop/dfs/data/current/subdir<Subdir Nr.>/<IP+Port>"*.

**J:** Messages about PacketResponders terminating, e.g., *" INFO dfs.DataNode$PacketResponder: PacketResponder <Responder ID> for block <Block ID> terminating"*. Each of the three clusters corresponds to one Responder ID (0, 1, 2). All three variants of the message use the same template, but do not appear interchangeably, causing them to be embedded slightly differently. Notably, each cluster can also contain embeddings of messages indicating the corresponding responder was interrupted, e.g., *"INFO dfs.DataNode$PacketResponder: PacketResponder <Responder ID> for block <Block ID> Interrupted."*. This message appears in normal and abnormal sequences and seems to appear interchangeably with the terminating message from the PacketResponder with the same ID.

**K:** Messages about java.io.IO Exceptions, e.g., *"INFO dfs.DataNode$DataXceiver: writeBlock <Block ID> received exception java.io.IOException: Could not read from stream"*. All messages from this cluster only appear in abnormal sequences.

**L+M:** Messages about PacketResponders receiving blocks with information about the block size and source, e.g., *"INFO dfs.DataNode$PacketResponder: Received block <Block ID> of size <Block Size> from /<IP>"*. While all the messages in this group use the same template, they are embedded into multiple distinct clusters. The clustering is based on the source IP address and the block size. For example, clusters in group L contain messages about blocks with a size of 67,108,864, while messages from clusters in group M reference smaller blocks.

**N:** Messages about the DataXceiver receiving blocks with information about source, destination and block size, e.g., *"INFO dfs.DataNode$DataXceiver: Received block <Block ID> src: /<IP+Port> dest: /<IP+Port> of size <Block Size>"*.

**O:** Messages about receiving a block, e.g., *"INFO dfs.DataNode$DataXceiver: Receiving block <Block ID> sre: /<IP+Port> dest: /<IP+Port> "*. These messages typically form the start of a log session.

**P:**Messages about transmitting blocks, e.g., *"INFO dfs.DataNode$DataTransfer. <IP+Port> :Transmitted block <Block ID> to /<IP+Port>"*. This group is split into two clusters, depending on the values of parameters. Most messages of this cluster originate from abnormal sequences.

**Q:** Messages of this group contain various rare exceptions and messages about metafile modifications. All messages only appear in abnormal sequences.

Summarizing the observations, the MessageEmbedder performs a function related to that of a log parser. Many of the clusters in the embedding space are formed by messages that share a common template. In some cases messages that might appear in the same context occupy the same region in the embedding space, e.g., messages about successfully served blocks and messages about exceptions while serving blocks. Other than parser based methods, ContraLog can embed messages of the same template differently depending on the values of parameters, e.g., different embeddings for different PacketResponders terminating.

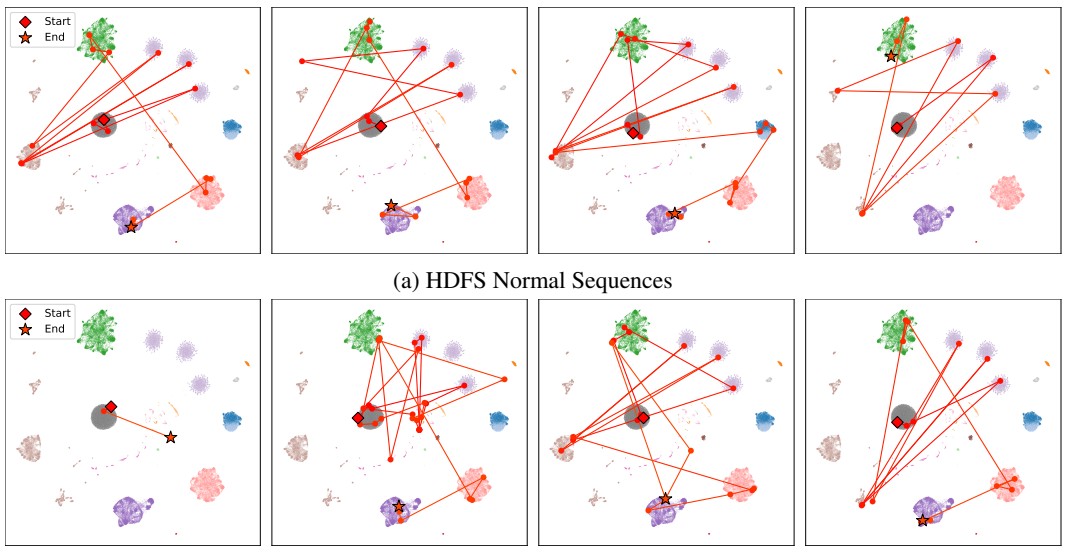

(a) HDFS Normal Sequences

(b) HDFS Abnormal Sequences

Figure 11: Comparison of sample normal and abnormal HDFS sequences in the embedding space. Each line visualizes the chronological order in which messages of a session were generated.

Figure 11 compares the sequences of normal and abnormal HDFS log messages embeddings. A typical normal sequence might look like this: A block is written to three DataNodes, creating three logs belonging to group O. Each PacketResponder terminates and confirms the transfer of the block (often 64MB), generating 6 messages, three from group J and three from group L. The NameSystem updates its block map with the new block locations and size, creating three messages of group E. Eventually, the NameSystem adds the blocks to an invalid set (three messages of group H) and they

are deleted (three messages of group I). Sequences can deviate from this pattern and still be normal, e.g., skip the declaration as invalid and deletion of a block, as shown in the fourth image of Figure 11a. Abnormal sequences often, but not always, deviate from this general pattern. An easy to spot example would be a very short session that ends abruptly with an exception, as shown in the first image of Figure 11b.

## A.10 IMPLEMENTATION DETAILS AND REPRODUCIBILITY

This section will give information about parameters used for training and testing. ContraLog was trained on a Quadro RTX 6000 with 24GB of VRAM. The memory requirements mainly depend on the model size, sequence and message length and batch size. The batch size was set to the largest value that fit within the memory constraints during training. Table 7 shows the hyperparameters used for the experimental evaluation. These parameters were mostly set to satisfy computational constraints. An extensive hyperparameter search was not conducted. The tokenizer vocabulary size was set to ensure the average log message can fit into the 64 token context window. Model parameters were optimized by AdamW with $\beta_1$ as 0.9, $\beta_2$ as 0.999, and a weight decay of 0.01. The parameter gradient norm ($\ell_2$) was clipped to a value of one. The temperature parameter $\tau$ was set to 0.25. Model training was done with the help of PyTorch. The overall parameter counts range from almost 500,000 to more than 4.5M, remaining well below the size of many transformer-based NLP models. Yet, due to the up to 16,000 cumulative tokens that can be contained in a single sequence, the memory requirements can grow for the Thunderbird dataset. The number of attention heads and layers for the MessageEncoder and the SequenceEncodes was set to be identical.

Table 7: ContraLog hyperparameters for different datasets

| Hyperparameter | HDFS | BGL | Thunderbird |
|---|---|---|---|
| max. message length (BPE tokens) | 64 | 64 | 64 |
| max. sequence length | 64 | 256 | 256 |
| tokenizer vocab size | 512 | 1024 | 2048 |
| BPE token embedding size | 128 | 64 | 128 |
| message embedding size | 512 | 256 | 512 |
| number of layers (both encoders) | 6 | 4 | 4 |
| number of heads (both encoders) | 6 | 4 | 4 |
| learning rate | 1e-4 | 1e-4 | 5e-5 |
| masking ratio (train) | 15% | 15% | 15% |
| batch size | 256 | 128 | 32 |

Training time varies between datasets. Using the described configuration, the HDFS model converges after 163 epochs, taking around 36 hours. On the BGL dataset loss converges at around 260 epochs (18 hours). Training on the full Thunderbird dataset poses much more of a challenge. To achieve the reported results, we trained for 6 epochs (36 hours). Mixed precision with autocast to 16-bit was applied to improve memory usage and computational efficiency.

Sequences are split 60%, 5%, 30%, 5% for training, validation, testing, and as a reference for calculating anomaly score distributions. Sequences are split in sequential order, meaning the newest sequences are used for testing and the oldest for training. This approach better simulates real-world scenarios and prevents data leakage. Abnormal sequences are discarded during the training, validation, and reference phases. For the test split, an equal ratio of normal and abnormal sequences is maintained. Normal sequences are randomly selected from the full test split, and any normal surplus sequences are discarded. For the HDFS dataset the timestamp of the first message in a session was used as a reference for the entire sequence. Table 8 summarizes the number of sequences in the final data splits.

Table 8: Number of sequences in the final data splits.

| Dataset | Train | Validation | Normal Test | Abnormal Test |
|---|---|---|---|---|
| Thunderbird | 467,483 | 38,857 | 31,624 | 31,624 |
| HDFS | 333,877 | 27,618 | 3,235 | 3,235 |
| BGL | 32,398 | 2,709 | 1,617 | 1,617 |

## A.11  SEQUENCE AND MESSAGE MODIFICATIONS

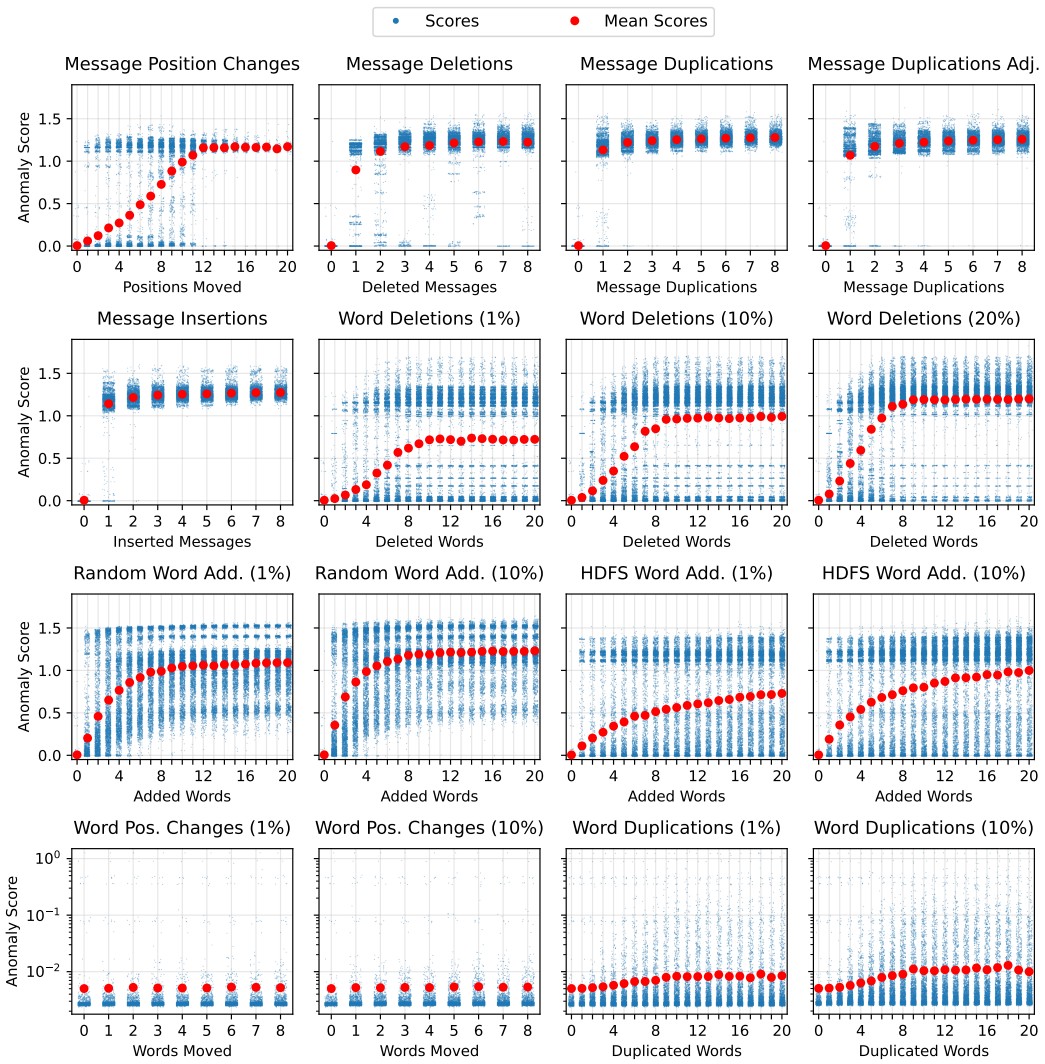

Figure 12: Anomaly score distributions of normal HDFS sequences after modifications on sequence and message-level.

Figure 12 shows the change in anomaly score when normal HDFS test sequences are modified. For these experiments, 2500 normal test sequences were used.

**Message Position Changes** shows the contextual anomaly score for sequences where the position of a single log messages in the sequence was changed by a certain number of positions. Zero position changes represents the baseline scores for the original unmodified sequences. The scores for most sequences remain low when a message is moved by just one position. However, moving some messages by just one position can cause a significant increase in the anomaly score.

**Message Deletions** shows the anomaly score for sequences with a certain number of messages removed. While the removal of some messages has little influence over the score, the deletion of most messages causes a large increase in the sequence anomaly score. Together with the previous experiment, this indicates that most messages in a sequence play an important role in identifying a sequence as normal, with some flexibility in the order they appear in.

**Message Duplications** and **Message Duplications Adj.**  show the anomaly scores for sequences where a certain number of random messages were duplicated. In the adjacent variant, the duplicate

Table 9: ContraLog F1 scores on HDFS for different levels of contamination.

| Contamination (%) | 0% | 1% | 2% | 3% | 4% | 5% |
|---|---|---|---|---|---|---|
| **Precision** | 93.88 | 93.70 | 92.58 | 84.62 | 75.28 | 64.36 |
| **Recall** | 74.57 | 70.91 | 69.76 | 65.02 | 62.14 | 60.03 |
| **F1-score** | 83.12 | 80.73 | 79.57 | 73.54 | 68.08 | 62.12 |

message was inserted directly after the original message. In the non-adjacent variant, the duplicate message was inserted at a random position in the sequence. The anomaly scores increase with the number of duplicated messages. As with the message deletions, average anomaly score increases quickly even with just a single duplicate message. Only few modified sequences remain with a low anomaly score. Inserting messages adjacent to the original message causes slightly smaller scores when just one or two messages are duplicated.

**Message Insertions** shows the anomaly scores when a certain number of random messages were inserted into the sequence at random positions. The inserted messages were sampled from the entire test set. Just like in the previous examples, the addition of just a single message increases the average anomaly score significantly.

The remaining experiments regard message-level modifications. For this, words in a message were defined as consecutive characters separated by whitespaces.

**Word Deletions** shows the anomaly scores for sequences where a certain number of words in a certain fraction of messages were deleted. The experiment was repeated with deletions in 1%, 10% and 20% of messages in a sequence. The deletion of a single crucial word from a single message can already increase the anomaly score. In other cases, the sequence score remains low even with multiple words removed. The deletion of words in a larger fraction of messages tends to increase the average anomaly score. Notably, at least one word per message was always kept, so that the message did not become empty.

**Random Word Add.** shows the anomaly scores when a certain number of random words were added to either 1% or 10% of messages in a sequence. The added words are randomly drawn from the 1,000 most common words in the English language. The majority of these words never appear in the HDFS dataset.

**HDFS Word Add.** also shows the anomaly scores when a certain number of words were added to either 1% or 10% of messages in a sequence. However, the words for this experiment were sampled from the HDFS test set, with a sampling probability proportional to the word frequency. Compared to the addition of random words, the anomaly scores increase more slowly with the number of added words. Notably, while the addition of many completely random words almost never results in a low anomaly score, even with 20 added words from the HDFS dataset, some sequences remain with a low anomaly score.

**Word Pos. Changes** show the anomaly scores for sequences where a certain number of words in a certain fraction of sequences (1%, 10%) were moved to a different position in the same message. While the average anomaly score for sequences with unaltered messages is 0.005030. Shuffling eight words in a message only increases the average score to 0.008245. This indicates that the MessageEmbedder is robust to changes in word order. We attribute this behavior to the fact that messages in the HDFS dataset can usually be identified just by the combination of tokens present in the message, not their order.

**Word Duplicate** shows the anomaly scores for sequences where a certain number of words in a certain fraction of messages (1%, 10%) were duplicated. The duplication of words has only a small influence on the anomaly score, even when multiple words are duplicated in multiple messages.

## A.12 CONTAMINATION OF TRAINING DATA

For training, ContraLog assumes exclusively normal log sequences. In practice, however, training data might be contaminated with some abnormal sequences. To evaluate the robustness of ContraLog to such contamination, we conduct experiments where the labels for a certain fraction of abnormal sequences is randomly flipped. Only the test set remains unaltered. Table 9 shows the

results for different levels of contamination on the HDFS dataset. Small amounts of contamination (up to 2%) lead to moderately worse performance. From 3% onward, the decline becomes more pronounced, with the F1-score dropping to 62.12% at 5% contamination. This experiment shows that ContraLog can deal with limited amounts of contamination and highlights the importance of data quality for reliable anomaly detection. In practice, this risk could also be decreases by hand-labelling a small set of anomalies for the optimization of hyperparameters like the anomaly detection threshold.

## A.13 SEQUENCE LENGTH ANALYSIS

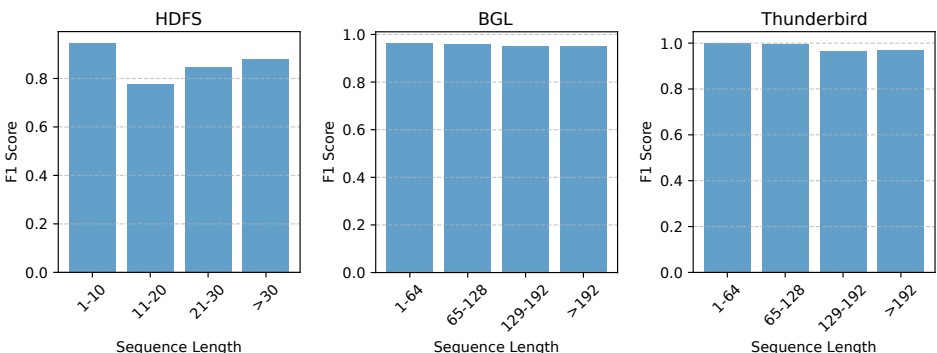

Figure 13: F1 scores achieved on on different test sequences grouped by sequence length.

Figure 13 shows the F1 scores achieved on different test sequences grouped by sequence length. The performance on the HDFS dataset is especially good for short sequences. This is likely because a short sequence length is a strong indicator of abnormal sessions that terminate prematurely. On the Thunderbird dataset, the performance only slightly decreases for longer sequences.

## A.14 THE USE OF LARGE LANGUAGE MODELS (LLMS)

For the writing of this paper, automatic tools were exclusively used to help with spell checking, grammar, and phrasing. All content is the original work of the authors.

## A.15 LIMITATIONS

**Dependence on Normal-Only Training Data**   Our approach assumes access to a sufficient amount of normal log sequences for self-supervised training. In practice, real-world log files are rarely labeled, and separating out purely normal data can be difficult. If abnormal entries leak into the training set, detection quality can decrease, potentially leading to higher false negative rates. Additionally, an underrepresentation of certain normal logs might introduce a bias and falsely flag them as abnormal. This might be the case for older/less common software or non-standard configurations.

**Computational Cost**   The hierarchical design of ContraLog reduces the sequence length for the transformers, but the quadratic complexity of self-attention in both MessageEncoder and SequenceEncoder still imposes limits on very long sequences. If an Anomaly only manifests over long time spans and a number of messages greater than the maximum context length, it is undetectable for ContraLog.

**Caching Memory Requirements**   During inference, we cache embeddings for repeated messages to avoid redundant computations. However, in environments with a large number of unique messages, the cache can grow large and consume significant memory. While this trade-off improves runtime performance, it may become impractical in some scenarios, unless additional pruning or eviction strategies are employed.

