# OpenReview forum: "ContraLog: Log File Anomaly Detection with Contrastive Learning and Masked Language Modeling"
_ICLR.cc/2026/Conference — Submitted to ICLR 2026_

### Official Review · Reviewer_Z3KG · 2025-10-31

**Soundness:** 2
**Presentation:** 3
**Contribution:** 2
**Rating:** 2
**Confidence:** 3

**Summary:**

This paper introduces ContraLog, a novel, self-supervised method for log anomaly detection that operates directly on raw log messages, eliminating the need for a separate log parsing step. The core idea is to reframe the problem from predicting discrete log template IDs to predicting continuous message embeddings.The paper's key contributions include :A parser-free and self-supervised framework for log anomaly detection; A two-pronged inference strategy that combines a contextual anomaly score (based on the discrepancy between predicted and actual message embeddings) and a point anomaly score (based on the deviation of a message's embedding from a set of normal embeddings).

**Strengths:**

1, The work is highly original. While the use of transformers and masked language modeling exists in prior log analysis works (e.g., LogBERT, LogFit), the core innovation lies in predicting continuous embeddings rather than discrete tokens. This elegantly circumvents the information loss inherent to log parsers. The creative combination of contrastive learning with masked language modeling for this specific task is novel and well-motivated.

2, The methodology is sound and well-explained, from the custom BPE tokenizer to the symmetric contrastive loss and the robust z-score aggregation for inference.

3, The paper is generally well-written and clear. The problem is effectively motivated, the architecture is illustrated (Figure 1), and the training and inference procedures are described in sufficient detail.

**Weaknesses:**

1, A significant weakness that must be addressed is the substantially lower performance reported for LogBERT and DeepLog compared to their original publications. While the authors provide a plausible justification (fitting the parser only on the training set and chronological splitting), the magnitude of the performance drop (e.g., LogBERT F1 of 55.51 vs. originally reported ~99+ on HDFS) is dramatic. This raises questions about the fairness of the comparison. A more rigorous experimental setup, such as reporting results with both the original parsing method and the proposed stricter method, is needed to conclusively demonstrate that ContraLog's gains are due to its architecture and not the experimental conditions.

2,The self-supervised approach crucially depends on the training data consisting purely of normal sequences. The paper correctly identifies this as a limitation, but could do more to quantify the method's sensitivity to contamination. An experiment injecting small, realistic rates of anomalous sequences into the training set would provide crucial insights into its robustness in noisy, real-world scenarios.

3,Although an ablation study is present, it is purely empirical (showing which score combinations work best). A more mechanistic ablation, such as training a version of ContraLog that predicts discrete template IDs (using the same architecture but a cross-entropy loss on parsed templates), would directly isolate the benefit of continuous embedding prediction versus the discrete approach. Furthermore, the excellent performance of point anomaly detection on BGL/Thunderbird, while explained, warrants a deeper investigation into why the contextual signal is less critical there.

**Questions:**

The results for LogBERT and DeepLog are far lower than those typically reported in the literature. Could you please provide a more detailed analysis to rule out implementation or experimental setup artifacts? For instance, have you attempted to replicate the original papers' parsing strategy (on the full dataset) to establish a performance upper-bound for these baselines under your evaluation split? A clear demonstration that the performance gap persists even when parsing is done "optimally" would significantly strengthen the claim that ContraLog's advantages are fundamental.

---

> ### Author Response · Authors · 2025-11-21
>
> We thank reviewer Z3KG for their feedback, questions, and acknowledgment of ContraLogs strengths.
>
> **Question 1 and Weakness 1:** LogBERT and DeepLog performance
>
> To address the concerns, we aim to provide LogBERT and DeepLog with conditions resembling those provided in the original LogBERT publication as closely as possible, while still maintaining the same data split.
>
> | | Precision | Recall | F1-score |
> |---|---|---|---|
> | HDFS - LogBERT | 98.80 | 64.90 | 78.34 |
> | BGL - LogBERT | 87.62 | 92.73 | 90.10 |
> | TBird - LogBERT | 90.56 | 94.86 | 92.66 |
> | HDFS - DeepLog | 92.21 | 64.05 | 75.59 |
> | BGL - DeepLog | 91.56 | 79.32 | 85.00 |
> | TBird - DeepLog | 89.78  | 94.02 | 91.85 |
>
>
> ~~Results for the Thunderbird dataset are still pending. We will report results as soon as possible and include them in Table 2.~~
> Differences from the previous setup:
>  - The parser is fitted on all logs before splitting. All templates not in the test set get mapped to the same out-of-vocabulary input.
>  - We discard sequences shorter than 10 messages - as in the original implementation.
>
> What still differs from the original conditions:
>  - We split data temporally, meaning only the earliest sequences are used for training. Originally, all normal sequences were shuffled and all abnormal sequences used in the test set [1].
>  - For Thunderbird and BGL: No overlap between moving log windows [1]. Each message can only be in one sequence and one subset.
>  - For Thunderbird: We work on the full 200m dataset, not the version with the first 20m messages that is often employed (sometimes called Thunderbird-mini).
>
> Despite not being able to reproduce the originally reported results (HDFS - LogBERT: 82.32 [2]) with our data split, we hope this reply can eliminate concerns regarding the implementation and evaluation of our baselines and give a better demonstration of ContraLog's benefits due to its architecture.
>
> **Weakness 2:** Contaminated training data
>
> We intend to perform the proposed experiment of training ContraLog with a portion of abnormal sequences in the training set. We will report results as soon as possible. As noted, this is a highly relevant issue for real-world applicability.
>
> **Weakness 3.2:** Effectiveness of point anomaly detection
>
> *Why does point anomaly detection work better for BGL and Thunderbird?* Point anomaly detection works well on these datasets, because they contain many such point anomalies. A similar observation was also made by Wittkopp et al. “[...] more than 99% of all anomalies in the Thunderbird and Spirit datasets are being classified as template anomalies.” [3].
> Figure 2 shows a partial separation between messages for normal and abnormal  BGL and Thunderbird sequences. Some clusters contain exclusively abnormal sequences. When a message from these clusters is processed, it is more easily identified as an anomaly. While exclusively abnormal messages do appear in the HDFS dataset, they are less numerous and appear in fewer sequences, which are often also identifiable by context anomaly detection.
>
> *Why does contextual anomaly detection not work well in BGL and Thunderbird?* HDFS sequences can be easily split (and are labeled) based on block IDs. This allows to group messages that are generated when processing the same data block. Figure 11 shows that normal HDFS sequences often consist of a certain number of messages using certain templates. Contextual anomaly detection is well suited for detecting sequences that deviate from this pattern.
> For the BGL and Thunderbird datasets, we were not able to describe similar simple patterns. The combination of messages generated by multiple processes and users makes it difficult to accurately predict a message based on the context. By splitting logs such that only closely related messages are contained in the same sequence, contextual anomaly detection might be improved.
>
> *Why can parser-based methods perform on BGL and Thunderbird when they do not perform explicit point anomaly detection?* Parsers serve as an implicit point anomaly detection. When encountering a message template not seen during fitting, it might be mapped to a mostly unrelated template ID, which is then unlikely to be predicted correctly. In doing so, parsers can, to some extent, help in detecting never before seen templates, but still fall short if point anomalies are abnormal because of log values and not the template.
>
>
>
> ---
>
> [1] Haixuan et al., Logbert: Log anomaly detection via bert. https://github.com/HelenGuohx/logbert/blob/main/BGL/data_process.py, accessed: 21 Nov. 2025
>
> [2] Haixuan, G., Shuhan, Y., Xintao, W. Logbert: Log anomaly detection via bert. In 2021 International Joint Conference on Neural Networks (IJCNN), pp. 1–8, 2021
>
> [3] Wittkopp, T., Wiesner, P., Scheinert, D., Kao, O. (2022). A Taxonomy of Anomalies in Log Data. In: Hacid, H., et al. Service-Oriented Computing – ICSOC 2021 Workshops. ICSOC 2021. Lecture Notes in Computer Science, vol 13236

---

> ### Author Response · Authors · 2025-11-27
>
> As suggested in **Weakness 2**, we investigated the effect that abnormal contamination can have on ContraLogs performance.
> For the experiments, we randomly add mislabelled abnormal sessions to all subsets, except the test set.
>
> | Contamination | 0% | 1%| 2%| 3% | 4% | 5% |
> |---|---|---|---|---|---|---|
> | Precision | 93.88 | 93.70 | 92.58 | 84.62 | 75.28 | 64.36 |
> | Recall| 74.57 | 70.91 | 69.76 | 65.02 | 62.14 | 60.03 |
> | F1-score  | 83.12 | 80.73 | 79.57 | 73.54 | 68.08 | 62.12 |
>
> Small amounts of contamination (up to 2%) lead to moderately worse performance. From 3% onward, the decline becomes more pronounced, with the F1-score dropping to 62.12% at 5% contamination.
>
> This experiment shows that ContraLog can deal with limited amounts of contamination and highlights the importance of data quality for reliable anomaly detection.
> In practice, this risk could also be decreases by hand-labelling a small set of anomalies for the optimization of hyperparameters like the anomaly detection threshold.

---

### Official Review · Reviewer_fatz · 2025-10-31

**Soundness:** 3
**Presentation:** 3
**Contribution:** 2
**Rating:** 4
**Confidence:** 4

**Summary:**

The paper proposes a self-supervised log file anomaly detection method called ContraLog. It uses a message encoder to encode individual log messages and a sequence encoder to model log sequences. ContraLog categorizes anomalies into two types: point anomalies and contextual anomalies. During inference, each message in a sequence is masked sequentially, producing two corresponding anomaly scores for every message. Experiments on three datasets show that the F1 score of the proposed method is higher than the baselines.

**Strengths:**

W1. The core idea of shifting from discrete template prediction to continuous embedding prediction is a significant strength. This directly addresses the well-known limitations of parser-based methods, namely the loss of information from variable parameters and the inability to capture semantic similarity between different templates.
W2. Comparing actual and predicted values is a common approach in time-series anomaly detection. This work innovatively applies this method to anomaly detection in textual logs, representing a valuable contribution.
W3. The proposed method ContraLog has a higher F1 score than other methods.

**Weaknesses:**

S1. In Table 2, the F1-score calculations for the ContraLog model across the three datasets appear to be incorrect. For example, on the HDFS dataset, the precision is 85.52 and the recall is 83.58, yielding an F1-score of 84.54, whereas Table 2 reports 83.35. Similar discrepancies are found for the BGL and Thunderbird datasets. The authors should provide an explanation for these differences.
S2. The Introduction section states, “Parsing Errors: Parsers require dataset-specific rules and frequent updates as log schemas evolve.” However, the ContraLog method also relies heavily on the training dataset and therefore does not appear to address this issue.

**Questions:**

1. Please explain the discrepancies in the F1-score values reported in Table 2.
2. Can the ContraLog method be transferred to log datasets that differ from the training dataset, for example by fine-tuning with a small amount of data or using a few-shot approach?

---

> ### Author Response · Authors · 2025-11-20
>
> Thank you for carefully reviewing the submission.
>
> **Question 1:** Precision, Recall, and F1-Score
>
> After investigating, we found that the reported results are unfortunately not the binary ones for the anomaly class, but the macro averaged scores for both classes. The corrected metrics are:
> | | HDFS| BGL| Thunderbird |
> |---|---|---|---|
> | Precision |93.88 |  94.68    | 95.03  |
> | Recall|74.57 | 99.13  | 99.84 |
> | F1-score  | 83.12   | 96.86  | 97.38 |
>
> The F1-score for HDFs and Thunderbird are slightly decreased (from 83.35 and 97.62). For BGL the binary score is greater (from 96.47). We also updated results reported in the ablation study and the threshold analysis.
> In the ablation study, the difference between features that work well and features that do not work well is more pronounced now. In the threshold analysis, the achievable score now drops further when the threshold approaches the 100th percentile. Other parts of the paper are not affected.
> Overall, we believe that this adjustment only has a minor impact on the main findings of the paper. Nevertheless, we sincerely thank you for highlighting this discrepancy, giving us the opportunity to correct it.
>
> **Question 2:** Fine-Tuning
>
> While not part of our experiments, ContraLog training can be started on one dataset and then continue on another.
> The outcome of fine-tuning will greatly depend on the data shift between both datasets. Log files can be highly unique to the specific software that created them. They often are repetitive and lack the semantic complexity of more general text data. Given this, patterns learned from one dataset are unlikely to transfer to an unrelated one.
> Given the low semantic variety, tokenizers fitted to one dataset will also poorly tokenize another. These are the results when using a tokenizer fitted on one dataset to tokenize the messages from another dataset:
>
> | Tokenizer | Data | Avg. tokens |
> |---|---|---|
> | HDFS | HDFS | 20.81 |
> | BGL | BGL | 24.88 |
> | HDFS | BGL | 53.96 |
> | BGL | HDFS | 54.04 |
>
> Both issues could be mitigated by relying on a general pre-fitted tokenizer or fitting on all datasets at the same time.
> For related datasets, like logs generated by the same software but running on different machines/locations, fine-tuning could be a useful tool.
>
> **Weakness 2:** Dataset specific rules
>
> By dataset-specific rules, we refer to the practice of many parsing approaches to apply specific regexes.
> With prior knowledge of the log structure, these can be used to remove or replace known log values with wildcards before the main parsing step.
> Examples we often see applied:
>  - (?<=blk_)[-\d]+    ---->      Block IDs (HDFS)
>  - (/[-\w]+)+    ---->     file paths
>  - (0x)[0-9a-fA-F]+    ---->      hexadecimals
>  - \d+\.\d+\.\d+\.\d+    ---->     IPs
>  - (?<=Warning: we failed to resolve data source name )[\w\s]+    ---->      data sources following a warning message (Thunderbird)

---

### Official Review · Reviewer_4dZc · 2025-11-02

**Soundness:** 2
**Presentation:** 2
**Contribution:** 2
**Rating:** 4
**Confidence:** 3

**Summary:**

The paper presents ContraLog, a method for log file anomaly detection that operates directly on raw log messages, avoiding the use of traditional log parsers. The core idea is to reframe the problem from predicting discrete log templates to predicting continuous message embeddings in a self-supervised manner. The model uses a hierarchical transformer architecture to learn embeddings for both individual messages and sequences of messages. Anomaly detection is based on a two-part scoring system: a contextual score, which measures how well a message embedding can be predicted from its surrounding context, and a point score, which measures the similarity of a message embedding to those seen in normal training data. The method's effectiveness is demonstrated on the HDFS, BGL, and Thunderbird benchmark datasets.

**Strengths:**

1. The central concept of predicting continuous embeddings instead of discrete log keys is a strong contribution. This directly addresses the information loss problem inherent in parser-based methods, which often discard important details contained in variable parameters.
2. The dual-pronged anomaly scoring mechanism is a practical and well-thought-out design. The point anomaly score provides a safety net for cases where contextual information is weak or misleading, such as a sequence of identical abnormal messages. The analysis in the appendix effectively shows how the importance of each score type varies by dataset, validating this design choice.
3. The evaluation is quite thorough. Beyond reporting standard metrics, the paper includes a qualitative analysis of the learned embedding space, which gives confidence that the model is learning semantically meaningful representations. This analysis shows the model can group related messages and distinguish between different parameter values for the same log template, supporting the main claims.

**Weaknesses:**

1. The "parser-free" claim feels a bit overstated. The method uses a custom Byte-Pair Encoding (BPE) tokenizer trained on each specific dataset. This is still a data-dependent preprocessing step that learns the statistical patterns of the log text, which is conceptually not so different from what a parser aims to achieve, albeit at a different level of granularity. "Template-free" might be a more accurate descriptor.
2. The training process seems to require a fair amount of dataset-specific tuning. As shown in Table 5, key hyperparameters like tokenizer vocabulary size, embedding dimension, and even learning rate were set differently for each dataset. The paper notes an extensive search was not conducted, but this raises questions about how to apply ContraLog to a new system. It's not clear how a practitioner would set these parameters without a labeled dataset for tuning.
3. The handling of long-range dependencies is a potential issue. The model's maximum sequence length was limited to 256 messages, and the authors acknowledge the quadratic complexity of transformers as a limitation. For anomalies that unfold over thousands of log messages, this architecture might not be suitable, and it's not clear how performance or computational costs would scale.

**Questions:**

1. Regarding the Point Anomaly Score: This score is calculated as the distance to the single closest embedding from a random subset of the training data (Section 3.3). This seems potentially sensitive to the choice of that subset. Were other, perhaps more robust, methods considered, such as using a k-NN distance or the distance to a cluster centroid of normal embeddings? Also, how was the size of this subset of normal training sequences determined?
2. On the HDFS Results: In Table 2, the statistical methods outperform ContraLog on the HDFS dataset. The paper suggests this is because sequential information is less important for HDFS anomalies (lines 425-428). However, the ablation study in Table 4 shows that using only the contextual scores (Cmax and Cmean) gives the best F1-score for HDFS. This seems to suggest that context is, in fact, the most important signal for this dataset. Could the authors clarify this apparent contradiction?
3. On the Practicality of the Threshold: The detection threshold is set at the 95th percentile of scores from a normal calibration set (line 330). In a production environment, this would imply a static 5% false positive rate on normal data, which could be too high. It would be helpful to see a Precision-Recall curve or similar analysis to understand the trade-off between flagging true anomalies and raising false alarms at different threshold settings.
4. Inference Latency: The paper mentions that caching embeddings for repeated messages can significantly reduce redundant computations (lines 342-346), noting an 89.1% reduction in embedding steps for Thunderbird. This is a great practical optimization. Could the authors provide some concrete numbers on the wall-clock inference time (e.g., latency per sequence) with and without this caching enabled? This would offer a clearer picture of the model's real-world performance.

---

> ### Author Response · Authors · 2025-11-20
>
> We thank reviewer 4dZc for their valuable and detailed feedback.
>
> **Question 1:** Point anomaly detection alternatives
>
> We did consider alternatives such as K-NN, but did not observe any benefits. Since our training data consists exclusively of normal data, we assume that a single sufficiently similar message is enough to rule out a point anomaly. This approach also allows us to deduplicate the reference set without losing performance.
> To demonstrate the influence of the reference set size on the performance, we conducted an additional experiment. Appendix A.8 discusses how the number of reference sequences influences point anomaly detection. Initially, increasing the number of references improves performance. Eventually, the embedding space of normal messages is sufficiently covered, and the score converges.
> Similar to this experiment, the number of reference sequences can be increased until the average minimum distance to messages from the evaluation set starts showing only marginal improvement. Eventually, adding more references becomes a performance trade-off with diminishing returns.
>
> **Question 2:** Context importance
>
> Statistical methods outperform LogBERT and DeepLog on HDFS.
> The message type count vectorization for statistical baselines discards the ordering of messages but still retains information about how often each message type appears in a sequence.
> For the HDFS dataset, we observe that the occurrence of message types in a sequence plays a more important role than the ordering.
> While the content of a normal sequence can vary, many follow a fixed format (see Figure 11a). The presence of the correct number of message types (e.g., 3x "Reciving Block", 3x "PacketResponder terminating", ...) can be an indicator of a normal sequence. Sequences that are missing critical messages or contain too many of one type can be identified without knowledge of the message order.
> This assumption is also supported by our synthetic anomaly experiments (see Appendix A.11). There, we found that removing a message from a sequence leads to a greater increase in the anomaly score than changing the position of a message in a sequence.
>
> **Question 3:** 95th percentile causes 5% false positives
>
> We agree that the chosen threshold has a major influence on ContraLogs performance. For our approach, an assumption about the expected contamination is required. To not rely on labeled anomalies, we heuristically choose the 95th percentile (see Figure 4).
> As suggested, we added precision-recall curves for each dataset (see Figure 5).
> The unique shape of the curve for HDFS is caused by the fact that a subset of anomalies is easy to spot (e.g., short/long sequences, exceptions that only occur in abnormal sessions). This makes it possible to maintain a low false positive rate even with low thresholds.
>
> **Question 4:** Inference latency
>
> Thank you for acknowledging the caching mechanism.
> We did perform the proposed experiment. Using an unlimited cache on the test sets, we can report the following results:
>  - Without cache: 0.0447s/sequence
>  - With 100% hit rate: 0.0277s/sequence
>  - The improvement when starting with an empty cache is now presented in Figure 8. As the cache fills, the average wall-clock time is reduced gradually, but varies depending on the specific content of a sequence.
>
> **Weakness 1:** “Parser-free” claim
>
> We acknowledge that our tokenizer performs a task related to that of a parser. We will attempt to tone down the "parser-fee" claims.
>
> **Weakness 2:** Hyperparameters
>
> As noted, no extensive hyperparameter search was conducted. We can, however, provide some qualitative insights for parameters.
> Vocabulary size: Chosen such that the average message fits comfortably in the MessageEncoder context window.
> Tunderbird learning rate: Is lower than the rest due to the smaller batch size, which was the result of computational constraints.
> For a systematic tuning of hyperparameters without labeled data, the anomaly scores on the normal validation set can serve as a stand-in for the anomaly detection performance.
>
> **Weakness 3:** Sequence length scaling
>
> The relation between performance and sequence length depends on the data. For the HDFS dataset specifically, we found that an unusually long session length is often a strong indicator of an abnormal sequence, causing ContraLog to perform particularly well on long sequences.
> Depending on the source of the data, logs from multiple largely independent systems might be aggregated into one stream. Since the context given by logs from an unrelated subsystem is unlikely to be relevant for the evaluation of a log from another, we expect a benefit from grouping messages by source component for complex systems. This would also increase the effective timespan a single log window could cover.

---

### Official Review · Reviewer_QeV3 · 2025-11-04

**Soundness:** 3
**Presentation:** 3
**Contribution:** 3
**Rating:** 6
**Confidence:** 3

**Summary:**

This paper introduces a novel  parser-free and self-supervised method for log file anomaly detection named ContraLog, which reframes log anomaly detection as predicting continuous message embeddings rather than widely-used discrete template IDs. ContraLog combines a message encoder that produces rich embeddings for individual log messages with a sequence encoder to model temporal dependencies across sequences. Masked language modeling and contrastive learning are adopted in the training process to predict masked message embeddings based on the surrounding context. Experiments on three benchmark datasets empirically demonstrate the effectiveness of ContraLogs on complex datasets with diverse log messages.

The major contribution of the paper is concluded as following:
1. ContraLog is a novel framework that combines self-supervised contrastive learning and masked language modeling for log anomaly detection, eliminating the need for a log parser by working directly on raw log messages. It also does not rely on a large amount of manually labeled log messages.

2. ContraLog includes a novel anomaly scoring mechanism that consists of a contextual anomaly score and a point anomaly score.
The contextual anomaly score measures the model's ability to predict an embedding based on its context, while the point anomaly score measures how much a single message's embedding deviates from those seen in the training set.

3. Empirical validation on three public log datasets demonstrates that ContraLog outperforms existing methods like LogBERT and DeepLog, showing robust performance in complex logging scenarios. Additional experimental results are reported in the appendix.

**Strengths:**

1. The novelty of the proposed method is good. The core idea of this research is reframing log anomaly detection from predicting discrete tokens to predicting continuous embeddings of raw messages, which distinguishes ContraLog from existing approaches. The combination of masked language modeling and contrastive learning is also novel and effective.

2. The paper is clearly written and well structured. The research motivation is well explained in the introduction and related work section. The description of the models, training objective, and inference process is well expained and easy to follow. Figure 1 is a concise and well illustrated summarization of the proposed method.

3. The experimental results of ContraLog has a significant advantage in terms of F1-score,. Specifically, the F1-scores on BGL and Thunderbird have reached >96%.

**Weaknesses:**

1. The contents about the masked language modeling part is not very clear. Please see the questions below. The sequence encoder part in section 3.1 seems that it takes all the message representations as input, but only the masked messages are used for computing the contrastive loss.

2. The experiment section in the formal contents is very short, and only contains a single group of results. I think an ablation study about the essential parts of the proposed method like message masking and contrastive learning should be  conducted to validated the effectiveness of these components.

3. As listed in the limitation section of the appendix, the proposed method relies on a large amount of normal log messages. If the training set contains many abnormal messages, the performance would deteriorate. The proposed method does not make fully use of labeled abnormal messages, which may be expensive to obtain and contain rich information.

**Questions:**

I have not fully understood the masked language modeling part of ContraLog, so I raise the following questions:

1. How is masked messages generated? I thought that a number of randomly selected log messages in a sequence is masked.

2. Is the contrastive loss only computed with the masked input messages? If it is, why don't we mask the messages before inputting them into the message encoder?

---

> ### Author Response · Authors · 2025-11-20
>
> We thank the reviewer QeV3 for their constructive feedback and questions.
>
> **Question 1:** How are masked messages generated?
>
> The masking approach differs between training and inference.
> During training, we randomly select multiple messages in a sequence to be masked. This allows the SequenceEncoder to predict multiple masked representations in one forward pass. During inference at each forward pass, only one message is masked. This is repeated for each message in a sequence before aggregating individual message anomaly scores to one score per sequence.
> In both cases, the embedding of the masked message is then replaced with a trainable mask-embedding (identical for all masked messages).
>
> **Question 2:** Is the contrastive loss only computed with the masked input messages?
>
> The contrastive loss is calculated between the original embeddings produced by the MessageEncoder (hidden under the mask) and the corresponding embeddings predicted by the Sequence encoder.
> Messages are not masked before the MessageEncoder so we can calculate the target embedding to be predicted for the masked messages based on the context given by the surrounding messages.
> In an extension to the described method, all remaining (not masked) message embeddings in a batch could serve as additional negative samples for contrastive learning. This would, however, increase the risk of sampling false negatives when identical messages are repeated in the same sequence.
>
> **Weakness 2:** Ablation study of essential components
>
> We acknowledge that an ablation study could provide additional insights. However, the masked language modeling and contrastive learning components are essential to the design of ContraLog and cannot be removed without fundamentally altering the method. We were however able to perform an ablation study on the point and contextual anomaly detection approaches (see Appendix A.7).
>
> **Weakness 3**: Use of labeled anomalies
>
> While the proposed method does not rely on labeled anomalies, if labeled data is available it can be used to tune the threshold. Appendix A.2 discusses the positive effect labeled anomalies can have on the performance of ContraLog, when used to tune the detection threshold.

---

### Author Response · Authors · 2025-12-02
**Summary of Main Manuscript Revisions**

Despite the changed review process, we thank all reviewers for their time and feedback, allowing us to improve the paper.
We have updated the manuscript and highlighted changes in blue.

Main changes and additions in the revised paper:
 - ContraLog Metrics (**fatz**): Table 2 now shows the binary F1-scores for ContraLog.
 - Baseline Metrics (**Z3KG**): Table 2 now shows LogBERT and DeepLog metrics under more favourable conditions (parser fitted on the full dataset, ...) for a strong upper-bound comparison. The original, stricter results are now placed in Appendix A.6.2.
 - Contamination Robustness(**Z3KG**): Added new experimental results showing model performance with contaminated training data in Appendix A.12.
 - Computational Efficiency (**4dZc**): Added an evaluation of the caching mechanism's wall-clock time benefits in Figure 7.
 - Point Anomaly Detection (**4dZc**): Added new experimental results showing point anomaly detection performance depending on the number of reference sequences in Appendix A.8.
 - ROC Curves (**4dZc**): Added ROC curves for all datasets in Appendix A.2.

---

### Meta-Review · Area_Chair_3uxZ · 2026-01-06

**Summary:**

Reviewers consistently recognized the novelty and importance of reframing log anomaly detection from discrete template prediction to continuous embedding prediction, and found the overall architecture and empirical results promising. However, several reviewers raised concerns about the reliability and stability of the experimental evaluation, including initially incorrect metric reporting, unusually weak baseline performance, and assumptions about parser fitting and thresholding that materially affected results. Although many of these issues were addressed during rebuttal, the scope of post-review corrections and protocol changes reduced confidence that the empirical evidence was fully mature and clearly supported by the originally submitted manuscript, which informed the final recommendation.

**Reviewer Concerns:**

The rebuttal addressed several important concerns by correcting metric reporting, clarifying the masked-embedding and contrastive learning procedure, providing fairer upper-bound baseline comparisons, and adding additional analyses on robustness, thresholding, and computational efficiency. However, concerns remain regarding the extent of post-review experimental changes, including revised metrics and baseline protocols, which raises uncertainty about the stability and readiness of the empirical results as originally submitted. As a result, while individual technical issues were clarified, confidence in the maturity of the overall experimental evaluation remains limited.

**Reviewer Scores:**

see above

---

### Decision · Program_Chairs · 2026-01-26

Reject